# Nondestructive flash cathode recycling

Weiyin Chen [1,6], Yi Cheng [1,6], Jinhang Chen [1,6], Ksenia V. Bets [2], Rodrigo V. Salvatierra[1], Chang Ge[3], John Tianci Li [1], Duy Xuan Luong[1,3], Carter Kittrell [1,4,5], Zicheng Wang[1], Emily A. McHugh[1], Guanhui Gao[2], Bing Deng [1], Yimo Han [2], Boris I. Yakobson [1,2,4] ✉ & James M. Tour [1,2,3,4,5] ✉

Effective recycling of end-of-life Li-ion batteries (LIBs) is essential due to continuous accumulation of battery waste and gradual depletion of battery metal resources. The present closed-loop solutions include destructive conversion to metal compounds, by destroying the entire three-dimensional morphology of the cathode through continuous thermal treatment or harsh wet extraction methods, and direct regeneration by lithium replenishment. Here, we report a solvent- and water-free flash Joule heating (FJH) method combined with magnetic separation to restore fresh cathodes from waste cathodes, followed by solid-state relithiation. The entire process is called flash recycling. This FJH method exhibits the merits of milliseconds of duration and high battery metal recovery yields of ~98%. After FJH, the cathodes reveal intact core structures with hierarchical features, implying the feasibility of their reconstituting into new cathodes. Relithiated cathodes are further used in LIBs, and show good electrochemical performance, comparable to new commercial counterparts. Life-cycle-analysis highlights that flash recycling has higher environmental and economic benefits over traditional destructive recycling processes.

The ever-increasing demand for portable electronic devices and electric vehicles has accelerated the production of commercial secondary batteries, especially Li-ion batteries (LIBs)[1–3]. The market for rechargeable LIBs reached ~$46 billion in 2022 and it is projected to be ~$190 billion in 2032, with a compound annual growth rate of ~15% (ref. 4). Furthermore, at the projected pace of Li and Co mining, the world's reserves of these battery metals will be unable to keep up with the demand by 2050 and 2030, respectively[5,6]. Since the expected life of most LIBs is <10 years, and often only 2 years[7], the foreseeable staggering accumulation of spent LIBs is disconcerting[8,9]. While the spent anode is mainly graphite and therefore less expensive and environmentally benign, the spent cathode consists of Li and other transition metals, accounting for ~35% of the total weight and ~45% of the cost of LIBs[7]. Therefore, an effective close-loop recycling of the spent cathodes is needed to

minimize the environmental release of these battery metals and to alleviate demand for remote mining[8,9].

Previous recycling methods can be mainly categorized into two strategies: destructive and nondestructive recycling approaches (Fig. 1a). These depend on whether the integrity of the three-dimensional cathode structure is retained after the treatments. Typical destructive routes, including pyrometallurgy[10,11], hydrometallurgy[12,13], and bio-metallurgy[14], can reclaim the simple metals, oxides, or their salts from the cathode waste (CW) by destructing the entire three-dimensional morphology of the cathode using some harsh conditions, such as prolonged furnace-heating temperatures or caustic reagents. Recently, a rapid thermal radiation method has also been reported to convert spent cathodes into metal/metal oxide core-shell catalysts[15]. Traditional destructive recycling strategies can involve complex procedures, consume considerable

[1]Department of Chemistry, Rice University, 6100 Main Street, Houston, TX 77005, USA. [2]Department of Materials Science and NanoEngineering, Rice University, 6100 Main Street, Houston, TX 77005, USA. [3]Applied Physics Program, Rice University, 6100 Main Street, Houston, TX 77005, USA. [4]Smalley-Curl Institute, Rice University, 6100 Main Street, Houston, TX 77005, USA. [5]NanoCarbon Center and the Rice Advanced Materials Institute, Rice University, 6100 Main Street, Houston, TX 77005, USA. [6]These authors contributed equally: Weiyin Chen, Yi Cheng, Jinhang Chen. ✉e-mail: biy@rice.edu; tour@rice.edu

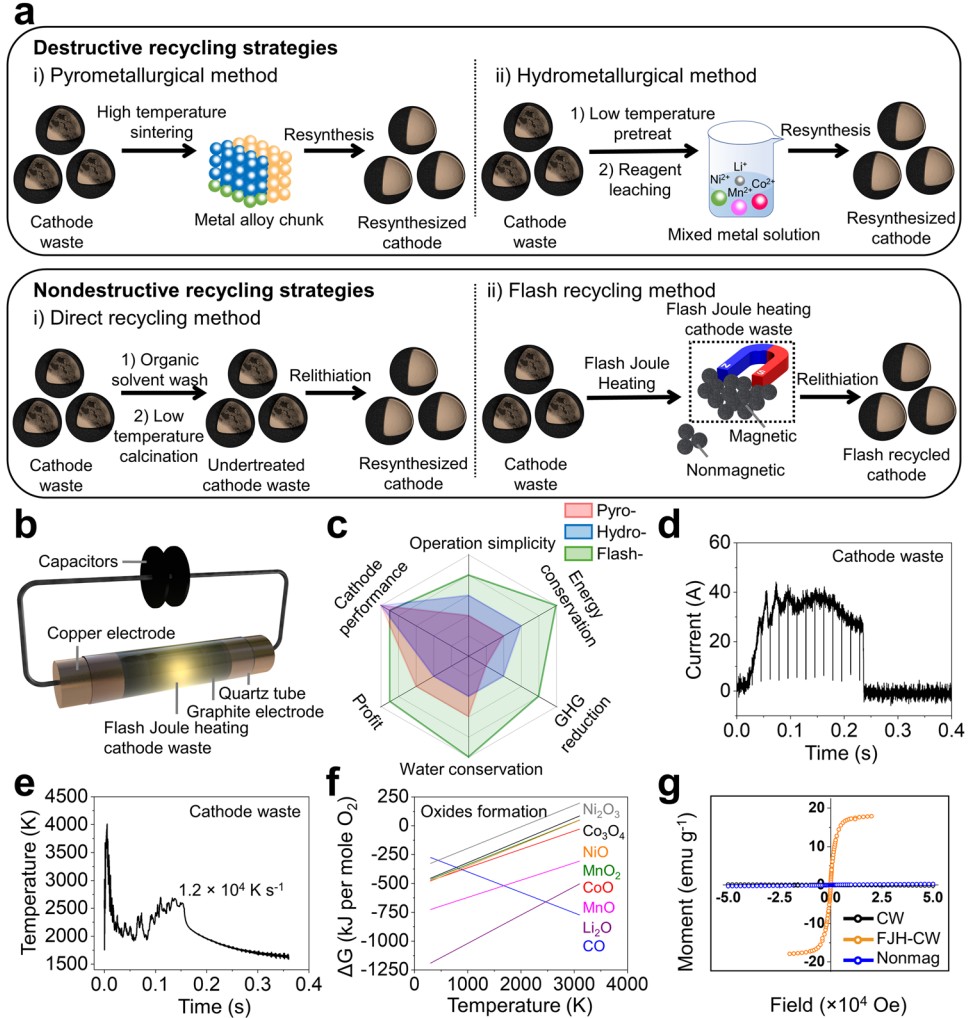

**Fig. 1 | Spend cathode recycling by rapid electrothermal process. a** Scheme about destructive and nondestructive recycling process, categorized by whether the integrity of the three-dimensional structure of the cathode is retained. The final resynthesis step is shown to highlight the individual precursors from each method. **b** The scheme about flash Joule heating process. **c** The radar plot related to comparison among different recycling strategies. **d** Current-time curve and **e** Real-time temperature measurement obtained from cathode waste. **f** Ellingham diagram of carbon monoxide and various metal oxides. **g** The magnetic response of cathode waste (CW, black), ferromagnetic portion of flash Joule heating cathode waste (FJH-CW, orange), and the non-ferromagnetic portion (nonmag, blue). CW: cathode waste. FJH-CW: flash Joule heating cathode waste. Hydro: hydrometallurgical method. Pyro: pyrometallurgical method. Flash: flash recycling method.

amounts of energy and water, and release much greenhouse gas (GHG) and secondary wastes (Supplementary Note 1), thereby increasing the cost of conversion back into the cathode morphologies[16].

The unique crystal structures of the cathode materials are essential, as are their chemical constitutions, as shown in the structure value estimation (Supplementary Table 1 and Supplementary Note 2)[17,18]. In addition, the common failure mechanisms related to the cathode materials include continuous electrolyte decomposition and cathode electrolyte interphase (CEI) accumulation, degradation of binder particles, loss of conductive contact between cathode particles, irreversible Li inventory loss, surface structure change, and dissolution of Cu and Al from current collector followed by migration towards electrodes[16,19]. Therefore, cathode healing through nondestructive strategies, focusing on how to effectively solve these failures, has recently gained more attention for battery recycling.

The direct recycling method is regarded as one of the nondestructive strategies since the bulk structure of cathode material is maintained during the whole process[20]. Typically, the CW is washed by organic solvents, such as $N$-methyl-2-pyrrolidone (NMP), followed by low-temperature calcination to remove binder and carbon additives. Then, the pretreated CW is used as the reaction precursors for subsequent cathode resynthesis (Fig. 1a). The common resynthesis methods mainly include solid-state calcination[21,22], hydrothermal regeneration[23,24], mechanochemical or electrochemical relithiation[3,25], molten salt repairing[26] and so on. Recently, an ultrafast repairing method using carbothermal shock method has also been proposed to replenish the Li inventory and reconstruct the surface structure[27]. However, the performance of the resynthesized cathode materials by direct recycling is sensitive to battery chemistries and states-of-health of the spent LIBs. The extent of surface structure degradation and impurity accumulation can lead to distinct electrochemical performance variations (Supplementary Table 2)[8–10,13,16,19–29] and abnormal capacity decay at the early stage for the resynthesized cathodes[9,27].

Here, we disclose a solvent- and water-free flash recycling method, including flash Joule heating (FJH) (Fig. 1b) combined with magnetic separation, followed by solid-state relithiation to recycle the untreated CW (Fig. 1a). The FJH process is ultrafast, and the particle structure is retained in the magnetic portion of the flashed product, called flash Joule heating cathode waste (FJH-CW). The other components, such as binder, CEI, conductive carbon, and metal impurities, are either decomposed or magnetically separated in the flash recycling method. This surface reaction facilitates the deagglomeration of the

**Table 1 | The flash conditions of different cathode materials**

| Reactant component | FJH-nLCO<br>90 wt% new LCO and 10 wt% CB | FJH-nNMC<br>90 wt% new NMC and 10 wt% CB | FJH-CW<br>80 wt% cathode waste and 20 wt% spent graphite |
|---|---|---|---|
| Mass | 200 mg per batch | 200 mg per batch | 150 mg per batch |
| Reaction atmosphere | Ar | Ar | Ar |
| Reactant resistance/ohm | 3 | 3 | 3 |
| Voltage/V | 120 | 120 | 150 |
| Reaction time/ms | 300 | 150 | 300 |
| Capacitance/mF | 60 | 60 | 60 |
| | **FJH-nLCO (gram-scale)** | **FJH-nNMC (gram-scale)** | **FJH-CW (gram-scale)** |
| **Reactant component** | **90 wt% new LCO and 10 wt% CB** | **90 wt% new NMC and 10 wt% CB** | **80 wt% cathode waste and 20 wt% spent graphite** |
| Mass | 800 mg per batch | 800 mg per batch | 600 mg per batch |
| Reaction atmosphere | Ar | Ar | Ar |
| Reactant resistance/ohm | 3 | 3 | 3 |
| Voltage/V | 120 | 120 | 150 |
| Reaction time/ms | 300 | 150 | 300 |
| Capacitance/mF | 132 | 132 | 132 |
| | **FJH-LCO** | **FJH-LCO (gram-scale)** | **FJH-NMC** |
| **Reactant component** | **80 wt% waste LCO and 20 wt% CB** | **80 wt% waste LCO and 20 wt% CB** | **80 wt% waste NMC and 20 wt% CB** |
| Mass | 150 mg per batch | 600 mg per batch | 150 mg per batch |
| Reaction atmosphere | Ar | Ar | Ar |
| Reactant resistance/ohm | 3 | 3 | 3 |
| Voltage/V | 150 | 150 | 150 |
| Reaction time/ms | 300 | 300 | 300 |
| Capacitance/mF | 60 | 132 | 60 |

used cathode particles, which is beneficial for solid-state relithiation. The resynthesized cathodes demonstrate comparable electrochemical performance as the new commercial cathode materials, proving the efficiency of the flash recycling process. Life cycle assessment (LCA) comparisons to the present destructive cathode recycling methods demonstrate that the flash recycling method avoids the use of caustic reagents, and significantly reduces the total energy and water consumption, contributing to lower greenhouse gas (GHG) emissions and operational cost, and therefore highlighting the economic and environmental advantages to recycle spent batteries through flash recycling (Fig. 1c).

## Results and discussion
### Spent cathode treatment by FJH process
In a typical FJH process (Fig. 1a), a mixture of cathode waste and conductive additives such as 10 wt% of carbon black or 20 wt% graphite from the anode waste, is slightly compressed inside a quartz tube between two electrodes[30–32]. Safety notes are listed in Supplementary Note 3. The capacitor banks in the circuit are used to provide electrothermal energy to the reactants for ~300 milliseconds (Supplementary Fig. 1). The carbon additives can surround the cathode particles, which not only bridges the inner circuit to increase the electrical conductivity of the mixture but also serves as the reductant for the cathode waste. Therefore, the current will mainly pass through the conductive carbon and the generated electrothermal energy will transfer from these hot spots to adjacent cathode particles, contributing to the local carbothermal reduction at the surface of cathode particles[33]. Since FJH is ultrafast, the momentary electrothermal process avoids most loss of volatile metals, such as Li, and preserves the particle morphology and their three-dimensional structure.

During the typical FJH process with a voltage of 150 V and a resistance of 3 Ω, the current passing through the sample is recorded to reach ~40 A in ~300 milliseconds discharge time (Fig. 1d). Therefore, the specific energy density is 0.31 kWh kg$^{-1}$ and specific input power reaches 3.73 kW g$^{-1}$. The temperature is measured through a 16-channel optical fiber spectrometer by black-body radiation fitting (Supplementary Fig. 2)[34]. The temperature is estimated to be ~2500 K and the ultrafast cooling rate is recorded at ~1.2 × 10$^4$ K s$^{-1}$ (Fig. 1e), confirming that the local carbothermal reduction can be thermodynamically favorable during the FJH process (Fig. 1f). For this work, we tested cathode wastes directly collected from spent LIBs, including LCO (LiCoO$_2$), NMC (LiNi$_x$Mn$_y$Co$_z$O$_2$, normally termed as NMC$xyz$) and their mixtures of LCO and NMC (Table 1).

The flashed product includes a mixture of the ferromagnetic portion (FJH-CW, ~90 wt%) and non-ferromagnetic portion (~10 wt%) (Fig. 1g and Supplementary Fig. 3). A simple magnet can be used to extract the desired ferromagnetic portion (Fig. 1a and Supplementary Fig. 4). The non-ferromagnetic portion can be collected and re-flashed for further recycling. The details are discussed in Supplementary Figs. 5, 6, and Supplementary Note 4. The generality of magnetic separation is demonstrated for common cathode chemistries, including LCO, NMC, and mixtures thereof (Supplementary Fig. 7), as it is found in commercial CW recovered from spent LIBs from discarded laptop computer batteries.

### Recovery efficiencies of various battery metals
At the high temperature of ~2500 K, the metal impurities were evaporated, while the other inert impurities such as binder and CEI, were decomposed, and most nonmagnetic conductive carbon was removed after the subsequent magnetic separation (Fig. 2a), as confirmed by the thermogravimetric analysis (TGA) results as shown in Supplementary Fig. 8. High recovery yields are essential for an effective close-loop recycling strategy[4]. The recovery efficiencies from various flashed products are quantified using inductively coupled plasma mass spectroscopy (ICP-MS). For FJH-LCO derived from waste LCO, the average recovery yields are ~94.2% for Co and ~96.3% for Li, respectively, (Fig. 2b) after just one flash treatment. To explore the relationship between the recovery yields and flash pulses for spent cathode with one single component, such as waste LCO, the battery metal contents within the nonmagnetic portion are measured (Supplementary Fig. 9).

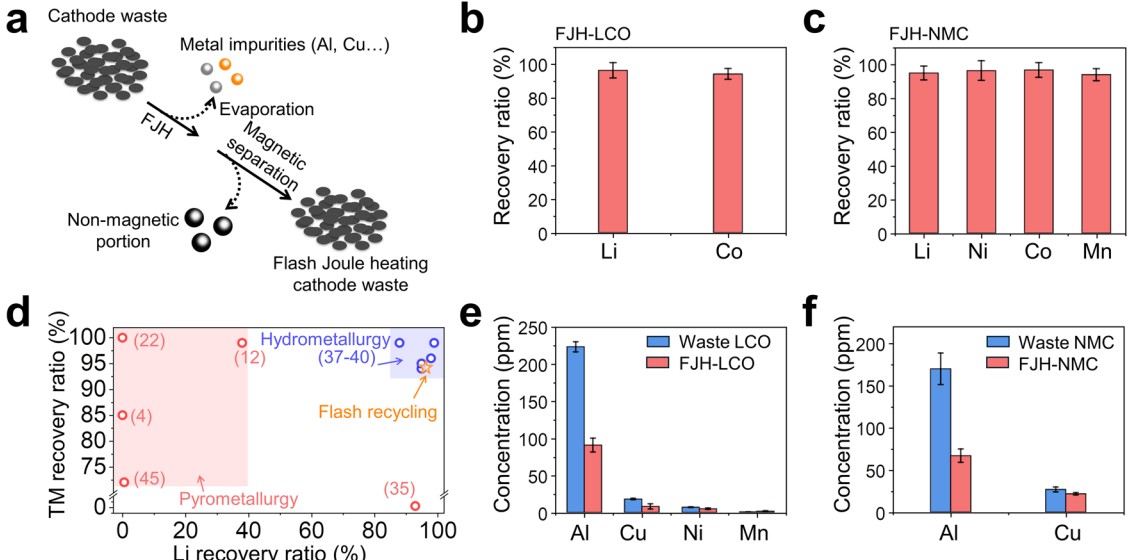

**Fig. 2 | Recovery efficiencies of various battery metals. a** Schematic showing the removal of common metal impurities by evaporation during rapid electrothermal process and the removal of other inert impurities by subsequent magnetic separation. **b** Recovery yields of Li and Co in the ferromagnetic portion of flash Joule heating LCO (FJH-LCO). The error bars reflect the standard deviations from at least three individual measurements. The same below. **c** Recovery yields of Li, Co, Ni, and Mn in the ferromagnetic portion of flash Joule heating NMC (FJH-NMC). **d** Comparison of recovery yields of Li and TM by different recycling methods, with references noted. **e** The concentration of impurity metals in waste LCO and FJH-LCO. **f** The concentration of impurity metals in waste NMC and FJH-NMC. TM: transition metals.

As the flash pulses increase from 1 to 3, the concentration of battery metals decreased from 1014.8 to 19.8 ppm for Co and from 99.8 to 11.1 ppm for Li, respectively, which indicated that the recovery yield for a single batch reaction increased marginally as the increase of flash pulses. Therefore, one flash treatment is used for spent cathodes with a single component in the LCA calculation. Similar high recovery yields are obtained for other cathode chemistries, like FJH-NMC, with the recovery yields of ~95.0%, 96.8%, 96.5% and 94.0% for Li, Co, Ni and Mn, respectively (Fig. 2c). Compared with conventional pyrometallurgical methods[4,35], a higher Li recovery yield can be achieved by the FJH process without compromising the recovery yield of Co (Fig. 2d). These values are also close to the leaching efficiencies by hydrometallurgical methods as shown in the blue region of Fig. 2d and Supplementary Table 3[4,11,13,35–45]. Therefore, when these flashed products are used as the reaction precursors for subsequent cathode resynthesis, the total amounts of the extra salts are limited, indicating a larger economic margin. The same tendencies were observed in FJH-CW derived from spent CW with mixed ingredients, including Li (92.7%), Co (93.2%), Ni (96.2%), and Mn (97.8%), as shown in Supplementary Fig. 10.

The purities of the products derived from different recycling methods are important since these products will be used as the precursors for subsequent cathode resynthesis (Fig. 2e, f)[46–48]. Recent work has demonstrated some metal impurities, such as $Cu^{2+}$ and $Al^{3+}$, can result in the decay of cathode performance by either reducing the actual ratio of active cathode materials for the inert impurities[9] or imparting electrochemically capacities for electrochemical active impurities[46,47]. This deleterious effect can be severe due to the gradual accumulation of impurities if multiple recycling loops are applied over the long term. It was reported that the Al and Cu contamination can induce the formation of secondary phase upon calcination, causing gradual specific capacity decay and polarization buildup at the cathode side[46,47]. In addition, these metal ions can be chemically reduced and subsequently deposited at the anode, which accelerates self-discharging and causes local dendrite formation[48]. Therefore, it is essential to control these impurities in the precursors, especially these metal contaminations within 100 ppm[9], which can be dissolved from the anode and cathode current collectors. However, there is no corresponding treatment to reduce the metal impurities for the direct recycling methods. For the waste LCO, the contents of Al and Cu are ~223.5 and ~18.2 ppm, respectively (Fig. 2e), while the concentration of these metals are ~90.9 and ~8.2 ppm, respectively for FJH-LCO, corresponding to notable content decreases by ~60% and ~55% for Al and Cu, respectively, indicating that FJH process can greatly reduce these metal contaminations. Since the total concentration of Al is low within the spent cathodes, there is no obvious formation of $LiAlO_2$ as confirmed by XRD results (Supplementary Fig. 11). These metal impurities can be further reduced after the solid-state relithiation step as shown in Supplementary Fig. 12. For flash-recycled LCO, the contents of Cu and Al contents are 4.6 and 75.1 ppm, respectively, indicating effective decreases of Cu and Al contents by ~75% and ~66%, respectively. A similar result is observed for FJH-NMC derived from waste NMC (Fig. 2f), where the concentrations of Al and Cu reduce from ~170.1 ppm and ~26.7 ppm to ~66.9 ppm and ~21.7 ppm, respectively, which are much lower than the safety content threshold of these impurities (~100 ppm, ref. 9). The removal of Cu and Al contamination can be achieved by the evaporation of these impurity metals during the momentary high temperature FJH process (Supplementary Fig. 13)[49–51], and subsequent magnetic separation since these contaminants are nonmagnetic.

## Morphology and structure of flash Joule heating products

To explore the evolution of cathode materials during the FJH process, we analyze the morphology of spent CW and FJH-CW by scanning electron microscopy (SEM) (Fig. 3a, b and Supplementary Fig. 14). There is no obvious bulk structure change, and both spent CW and FJH-CW show similar bimodal size distributions about the primary particles (Fig. 3c) where the peak at ~2 μm is from the waste NMC while the other peak is contributed by waste LCO. The same phenomena are observed for pure LCO (Supplementary Fig. 15) and NMC (Supplementary Fig. 16), and there is a similar particle size distribution before and after FJH treatment. These two cathode particles can be distinguished by X-ray diffraction (XRD) analysis as shown in Fig. 3d and Supplementary Fig. 11. The layered structure of

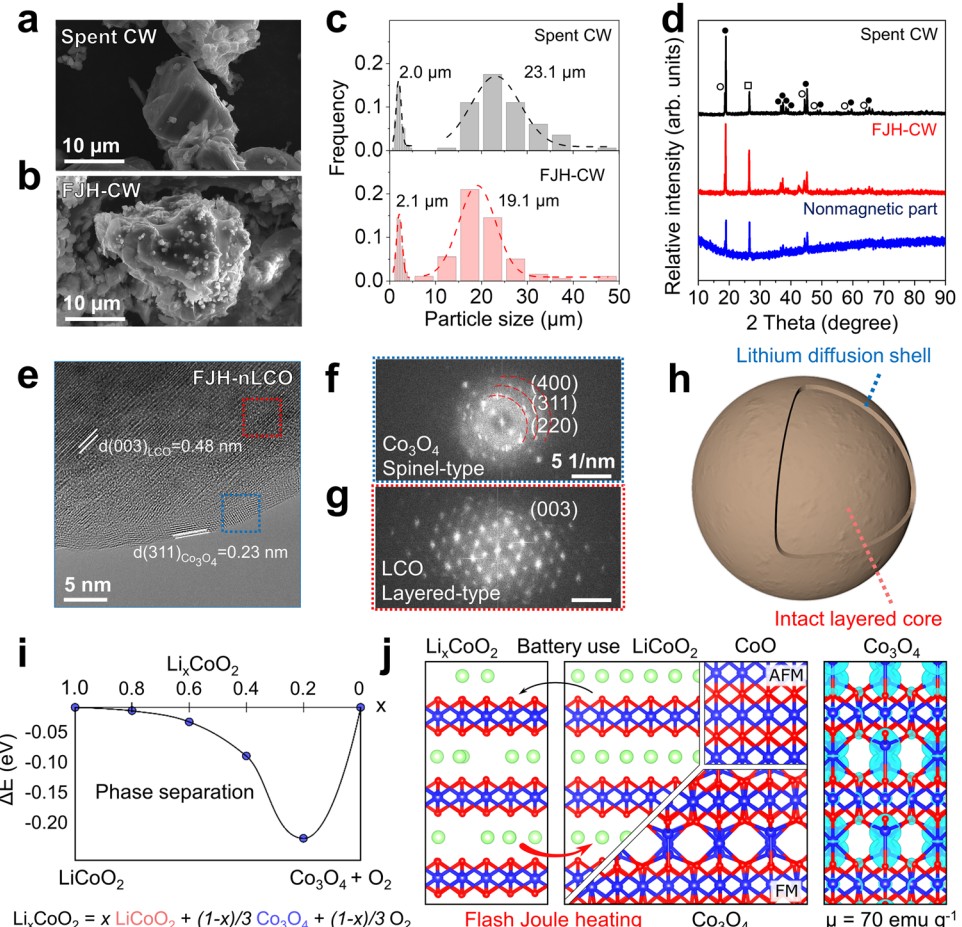

**Fig. 3 | Morphology and structure of flash Joule heating products.** SEM images of **a** spent CW comprised of waste LCO and waste NMC, and **b** FJH-CW. **c** The size distribution of primary particles from spent CW particles and FJH-CW. The number of particles $N = 200$. **d** The XRD spectra of spent CW (black), FJH-CW (red), and non-ferromagnetic flash product (blue). Square: Graphite. Hollow Circle: NMC. Solid circle: LCO. **e** HR-TEM images of the FJH-new LCO (nLCO) particles. **f, g** Fast Fourier transform results of FJH-nLCO particles from different areas in Fig. 3e. **h** The scheme showing the hierarchical structure of the FJH-LCO particles. **i** Computed energy preference towards phase segregation of spent LCO cathode. Adapted from previous work (ref. 9). **j** Atomistic structure of partially de-lithiated $Li_xCoO_2$ before FJH treatment, and high-quality $LiCoO_2$, $Co_3O_4$, and CoO obtained after FJH treatment. The right panel shows the magnetization of $Co_3O_4$ by plotting the computed spin polarization density isosurface at 0.02 e$^-$/Å$^3$ total magnetic moment of ~70 emu g$^{-1}$. FJH-CW: flash Joule heating cathode waste. FM: ferromagnetic. AFM: antiferromagnetic.

the FJH-CW is further evidenced by the (003) diffraction peaks of NMC and LCO at ~18.9° (ref. 28), indicating the existence of the intact core structure after FJH treatment. The nonmagnetic portion is mainly composed of the graphite conductive additive with some residual metal signals.

The surface morphology of the cathode particles is distinct before and after FJH treatment. For FJH-LCO particles, the formation of sub-microscale particles can be observed on the surface (Fig. 3b). To pin-point the surface structural change, high-resolution transmission electron microscopy (HR-TEM) is conducted for new LCO (nLCO) cathode after FJH treatment. For FJH-nLCO, there is a distinct shell structure with the thickness of ~2 nm, contributing from magnetic cobalt oxide (Fig. 3e). The fast Fourier transform (FFT) patterns confirm the presence of spinel type $Co_3O_4$ as shown in Fig. 3f. On the contrary, the layered core structure derived from the transition metal slab and Li slab can be distinguished beneath the shell. The corresponding FFT patterns at the core region confirm the cathode structure does not collapse after FJH treatment (Fig. 3g). The composition analysis of FJH-nLCO shows that there is a large binding energy shift towards higher value for Co 2p spectra at the surface of the cathode particles (Supplementary Figs. 17–18), indicating that the cathode surface was partially reduced from $Co^{3+}$ to $Co^{2+}$ and formation of

ferromagnetic oxides during the FJH treatment[9]. The formation of $Co_3O_4$ is also confirmed by the XRD result of FJH-nLCO as shown in Supplementary Fig. 17d. Similar phenomena on oxidation state reduction are observed for other cathode chemistries, such as new NMC (Supplementary Fig. 19). The splitting of the O 1s spectra to $O_\alpha$ (-532.6 eV) and $O_\beta$ (-530.3 eV), showed a transition from adsorbed oxygen species to lattice oxygen species (Supplementary Fig. 17)[52], which confirmed the existence of lithiated metal oxides below the surface. There was one extra peak resulting from C = O at the surface, which was not observed below the surface. Combined with the Li 1s spectra and FTIR results in Supplementary Fig. 6, it indicates that some lithium carbonate salt might form at the surface. Therefore, the appearance of magnetic properties underscores one of the most important aspects of the flash recycling method. The localized rapid heating and cooling can maintain the integrity of the particle while triggering a carbothermal reduction of cathode particles limited to the surface only (Fig. 3h). Based on the first-principles calculations results (Fig. 3i)[9], the structure change on the surface of the particle can be more thermodynamically favorable as the decay of states-of-health for spent cathode materials progresses resulting in a significant degree of delithiation (Fig. 3j). The as-formed reduction products include CoO and $Co_3O_4$ (Fig. 1f, Fig. 3j). The magnetic properties of a $Co_3O_4$/CoO

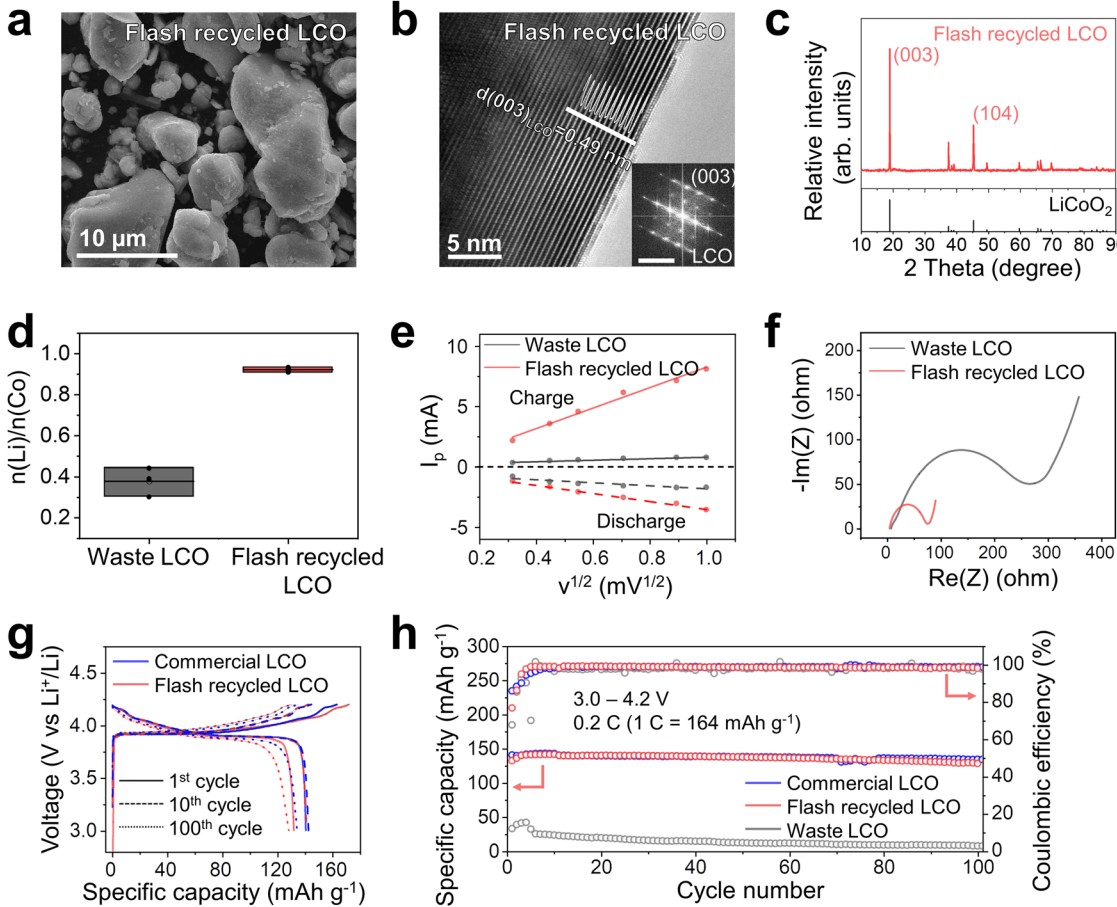

**Fig. 4 | Characterization and electrochemical performance of resynthesized cathode. a** SEM image of resynthesized cathode from FJH-LCO (flash-recycled LCO). **b** TEM image of flash-recycled LCO. The intensity profile in the image shows the alternative TM slab and Li slab, reflecting the layered cathode particles. The inset shows the fast Fourier transform results, confirming the layered structure. **c** The XRD spectrum of flash-recycled LCO. Powder diffraction file: 00-062-0420, LiCoO₂. **d** Molar ratio between cobalt and lithium from waste LCO and flash-recycled LCO. The center line and box limits of the plot represent the median and upper and lower quartiles, representatively. **e** Comparison of diffusion coefficients of Li⁺ in waste LCO and flash-recycled LCO. **f** Electrochemical impedance spectroscopy of waste LCO and flash-recycled LCO. **g** Voltage profiles of commercial LCO and flash-recycled LCO at different numbers of cycles. **h** Cycling performance of commercial LCO, flash-recycled LCO, and waste LCO with a Li anode at 0.2 C. $I_p$: peak current.

film are simulated (Fig. 3j, right panel), presenting a magnetic moment of ~70 emu g⁻¹ for bulk phase (magnetic moment approximately one-third of 219 emu g⁻¹ for Fe). The magnetic characteristics of oxide shells are partially reduced due to inherent disorder and size-effects displaying characteristics consistent with that of thin $Co_3O_4$ film[53-55]. The formation of this layer is critical for the subsequent magnetic separation. The gradual nature of annealing is consistent with the ability to improve recycling yield through the re-flash treatment of the nonmagnetic portion observed in the experiment. This simulation also indicates that FJH treatment can be more effective to treat heavily degraded cathode waste. The nanoscale structures of waste LCO and FJH-LCO are shown in Supplementary Fig. 20. Similar surface reconstruction to the spinel type $Co_3O_4$ can be observed for FJH-LCO derived from waste LCO. Besides, this FJH treatment can affect the CEI, binder, and electrolyte residue. Combined with magnetic separation, most of impurities can be removed from the cathode waste, guaranteeing the resynthesis of cathodes.

## Electrochemical performance of resynthesized cathode
The magnetic portion of the flashed product can be directly used for the subsequent solid-state resynthesis. Here, FJH-LCO is used as an example (Fig. 4a–c). The initial molar ratio between Li and Co is measured by ICP-MS to determine the extra amounts of Li₂CO₃, which will be mixed with FJH-LCO homogeneously for cathode relithiation.

For the waste LCO, this Li/Co ratio is ~0.4 (Fig. 4d), which is like the value in FJH-LCO (Fig. 2a). The ratio for resynthesized LCO is close to the stoichiometric ratio of pristine LCO, indicating effective lithium replenishment and cathode resynthesis. There is no obvious agglomeration of the resynthesized cathode materials derived from FJH-LCO, and each micro-grain-sized particle can be distinguished (Fig. 4a). This phenomenon is distinct from the resynthesized cathode prepared by the direct recycling method, where primary cathode particles stack together to form secondary particles with sizes up to ~100 μm (Supplementary Fig. 21), indicating that the flash recycling prevents agglomeration of resynthesized cathode materials. The existence of aggregates can prevent the full and close contact between the reaction precursors and lithium reagents during the resynthesis step, which can be another reason for the large performance difference. Furthermore, those nanoparticles formed at the surface of FJH-LCO disappear, reflecting the structure rearrangement after solid-state resynthesis (Fig. 4a). The size distribution is analyzed using SEM images (Supplementary Fig. 21) and the size below which 10%, 50%, and 90% of all particles are found (D10, D50, D90) are compared between commercial LCO and flash-recycled LCO (Supplementary Fig. 22). D10, D50 and D90 are 6.1, 13.6 and 19.7 μm, respectively, for flash-recycled LCO, while these values are 5.7, 12.5 and 19.8 μm, respectively, for commercial LCO. Therefore, commercial and flash-recycled LCO have similar size distribution.

The HR-TEM image and corresponding FFT patterns also confirm the formation of layered structure at the surface with high crystallinity, and the intensity profile reflects the alternating Co slab and Li slab (Fig. 4b). This result is consistent with XRD analysis as shown in Fig. 4c, where highly crystalline LCO with the space group R3̄m is shown. In the XRD results, (003) peaks indicate the property of layered structure in lithiated metal oxides, and (104) peaks reflect the property of transition metal-oxygen bond basic units which forms the layered compounds. The intensity ratio between (003) and (104) peaks is defined as structure factor in the work, indicating the efficiency of crystallization. The lower value of the structure factor reflects the cation mixing between the transition metal and lithium and generally a decomposition of the layered character[20]. This value is ~2.8 for the resynthesized LCO from the flash recycling method, which is much higher than the ~1.7 for regenerated LCO from the direct recycling method and is comparable to the value of ~3.0 for new commercial LCO (Supplementary Fig. 23). A summary of the structure evolution from waste LCO, to FJH-LCO and final flash-recycled LCO is shown in Supplementary Fig. 24, which highlights the surface reaction features of FJH treatment and cathode structure reconstruction after the solid-state reaction.

The LCO from flash recycling shows negligible mass loss (~0.25 wt %) from 298 to 823 K (Supplementary Fig. 8). The other magnetic impurity contents are also analyzed using a magnetic property measurement system (MPMS) as shown in Supplementary Fig. 25, since the magnetic impurities can accelerate the self-discharge of the cathode materials[56]. Based on the magnetic moment without external field, the total magnetic content of flash-recycled LCO is estimated to be ~94 ppb. This value is close to the industrial standard of <50 ppb for battery-grade materials and further optimization can be helpful to meet the standard during the industrialization and commercialization stage. The magnet test also confirms that no cathode powder can be attracted (Supplementary Video 1) as suggested by the previous literature[56]. Considering all the impurities, as characterized by TGA, ICP-MS, and MPMS, the purities of flash-recycled LCO is ~99.74%, which meets the standard for battery-grade cathode materials of >99.5%. The TGA results show no obvious mass loss (~0.28 wt%) until 1273 K, which reflects its high thermal stability and is comparable to the new commercial LCO (Supplementary Fig. 8).

The electrochemical properties are tested in coin cells for different cathode materials. The diffusion coefficient of $Li^+$ can be calculated by changing the scanning rates from 0.1 to 1.0 mV s$^{-1}$ in cyclic voltammetry (CV) analyses (Supplementary Fig. 26)[19]. For the fitting results between the peak current ($i_p$) and scanning rate ($v^{1/2}$) as shown in Fig. 4e, the diffusion coefficient of flash-recycled LCO is recovered from waste LCO, and the value is ~1.8× higher compared to direct recycling LCO during the charging stage (Supplementary Fig. 26d), presumably arising from the complete deagglomeration of the resynthesized cathode particles and better crystallinity with less cation mixing for flash-recycled LCO. There is also a smaller interfacial polarization and charge transfer resistance for flash-recycled LCO as calculated from electrochemical impedance spectroscopy (EIS) analysis, compared to waste LCO (Fig. 4f) and direct recycling LCO (Supplementary Fig. 27). Therefore, the flash-recycled LCO can deliver a discharge specific capacity of 142, 123, 99 and 56 mAh g$^{-1}$ at the cycling rate of 0.2 C, 0.4 C, 0.8 C, and 1.6 C, respectively (Supplementary Fig. 28).

The voltage profiles of flash-recycled LCO and commercial LCO at different cycle numbers are compared with the cycling rate of 0.2 C and areal capacity of ~1.5 mAh cm$^{-2}$ (Fig. 4g). Similar polarization curves during the charging and discharging stage can be observed. The overpotential is also much smaller than waste LCO when cycling at the same rate (Supplementary Fig. 29). After 100 cycles, the specific capacity of flash-recycled LCO is ~128 mAh g$^{-1}$, with the capacity retention of ~96.9% (Fig. 4h, Supplementary Fig. 30a), demonstrating

great improvement after the flash recycling process. The specific capacity of commercial LCO is ~134 mAh g$^{-1}$, with the capacity retention of ~95.5% after 100 cycles, which indicates that flash-recycled LCO can achieve a similar electrochemical cycling stability. The electrochemical performance of resynthesized LCO using commercially viable recycling methods, such as hydrometallurgical method[9] is shown in Supplementary Fig. 30, and similar specific capacity can be obtained. Therefore, the proposed flash recycling method is effective to regenerate the cathode materials from cathode waste.

**Economic and environmental analysis of flash recycling method**

Using GREET 2020 and EverBatt 2020 software packages developed by Argonne National Laboratory for determining the closed-loop life cycle analysis of LIBs[17,18], we compared the efficiencies of flash recycling with different types of recycling processes, including the hydrometallurgical (Fig. 5a), pyrometallurgical (Fig. 5b), flash recycling (Fig. 5c) and direct recycling methods (Supplementary Note 1 and Supplementary Tables 4, 5). The prospective cradle-to-gate life cycle assessments (LCA) are applied, which consists of the processes from the collection of individual intermediates from ~1.00 kg spent LIBs (cradle) by different types of reactions to the production of ~0.35 kg cathode using these intermediates as the reaction precursors at the factory (gate). The usage of the cathode materials and their disposal (grave) are not considered in this part, since it is assumed that cathode materials produced by various methods have the same usage and recycling stages[57]. A more detailed discussion about LCA is shown in Supplementary Note 1. The life cycle inventories with detailed parameters regarding the inputs and outputs of each individual step for the above methods are listed in Supplementary Table 5.

The proposed flash recycling method belongs to the non-destructive recycling strategy, which does not consume extra chemical reagents, such as hydrochloric acid (Fig. 5d), or energy to destroy the intrinsic cathode structures. Thus, the cradle-to-gate LCA (Fig. 5d–h) reflects that the flash recycling method avoids the use of concentrated HCl and decreases the consumption of water and energy by ~83% and ~62%, respectively, compared to the hydrometallurgical method. Therefore, the GHG emissions and the cost can be reduced by ~72% and ~58%, respectively. These values are close to the direct recycling method, which is the other nondestructive recycling strategy for now. Larger improvement can be seen when comparing flash recycling with the pyrometallurgical method. The flash recycling method reduces water consumption by ~80%, energy consumption by ~67% and GHG emissions by ~82%, reflecting the decrease in the environmental footprint and leading to decrease in estimated cost by ~41% (Fig. 5d–h) compared to the pyrometallurgical method. With increased interest in cathode materials possessing low Co content[58,59], such as NMC622, NMC811, and NCA[60,61], and non-Co-based systems[62–64] such as LiFePO$_4$ and LiNiO$_2$, more efficient recycling with increased profit margins might be attainable by flash recycling using these ferromagnetic metals.

To demonstrate the potential scalability of flash recycling method, the gram-scale FJH experiments are carried out for different cathode materials. The parameters can be seen in Table 1. There are several general strategies to maintain the specific energy density during the FJH treatment, including increasing the capacitance, flash repetitions, and voltages[9]. The final flashed products are shown in Supplementary Fig. 31. Since laboratory kilogram-scale graphene production has been achieved using an automated system with a more demanding condition, including higher temperature of >3500 K and longer duration of several seconds[65], the FJH process can presumably be integrated into a similar continuous system for spent LIB recycling. In addition, recent works have demonstrated that the same FJH process and carbothermal shock method can be used to achieve the effective regeneration of the graphite anode from spent graphite[33,49,66,67], indicating that flash recycling method can

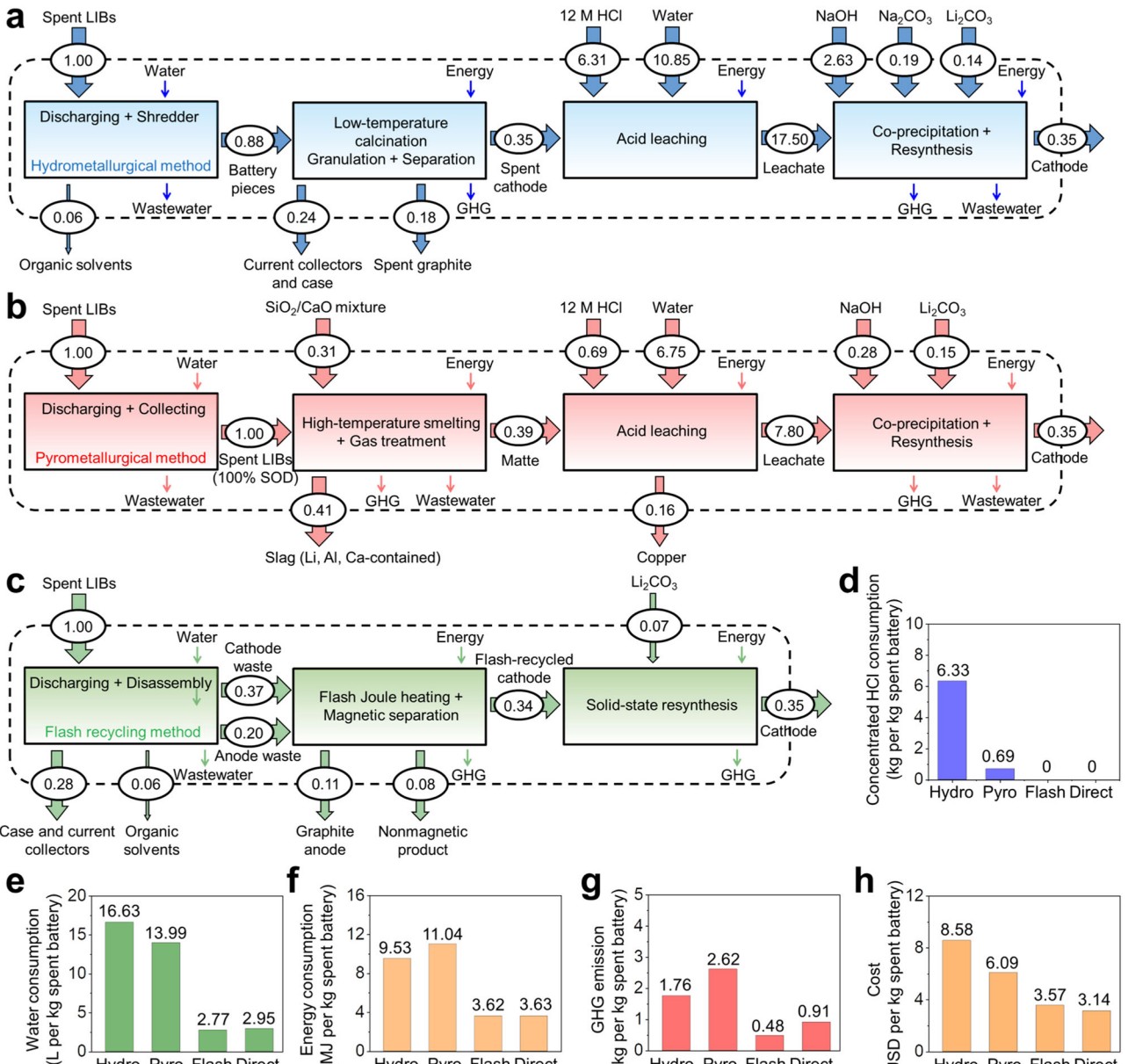

**Fig. 5 | Economic and environmental analysis of flash recycling method.**
**a**–**c** Process flow diagrams of various spent lithium-ion battery recycling routes, displaying the life cycle inventory including all considered inputs and outputs. Incidental inputs and outputs are shown with small arrows to differentiate them from explicit inputs and outputs. **a** Hydrometallurgical method.
**b** Pyrometallurgical method. **c** Flash recycling method. The unit is kg for all material flow. **d** Concentrated 12 M HCl consumption in treating 1 kg of spent batteries. **e**–**h** Water consumption, energy consumption, greenhouse gas emission, and cost analysis in treating 1 kg of spent batteries followed by producing -0.35 kg cathode materials from their individual precursors. Hydro: hydrometallurgical method. Pyro: pyrometallurgical method. Flash: flash recycling method. Direct: direct recycling method. GHG: greenhouse gas.

simultaneously solve both cathode and anode recycling problems arising from spent battery accumulation. Finally, since the FJH process is being industrially scaled to 1 ton per day scale per facility[68], manufacturability is attainable while minimizing dependence on freshly mined metal ores to produce LIBs.

## Methods
### Materials
The lithium cobalt oxide (LCO, 99.8% trace metals basis, 442704-100G-A) was purchased from Millipore−Sigma. Cathode nickel-manganese-cobalt (NMC, EQ-Lib-LNCM811) powder was purchased from MTI Corporation. Several different spent commercial lithium batteries were used for flash recycling, including battery-1 (LG Chem 112711, B052R785-9005A) obtained from Lenovo laptop computers, and battery-2 (18650-cylinder cells, LGDAHB21865-P308K034A3) obtained from local recycler at Houston, Texas. Carbon black (CB, APS 10 nm, Black Pearls 2000) was purchased from Cabot Corporation. Quartz tubing (*ID* = 4 mm, *L* = 6 cm) was used as the reactant FJH tube for small batches (200 mg per batch) and quartz tube (*ID* = 8 mm, *L* = 6 cm) was used for larger batches (800 mg per batch) in the experiments. The standard solutions for ICP tests included cobalt standard (1000 ± 2 mg/L, 30329-100ML-F), lithium standard (998 ± 4 mg/L, 12292-100 ML), manganese standard (1003 ± 5 mg/L, 74128-100 ML), and nickel standard (998 ± 4 mg/L, 28944-100ML-F), all of which were purchased from Millipore−Sigma. The nitric acid (HNO₃, trace metal grade, 1120060) was purchased from Fisher Chemical and hydrochloric acid (HCl, 99.999% trace metals basis, 339253-100 ML) was purchased from Millipore−Sigma. Water (HPLC Plus, 34877-4 L) was

purchased from Millipore–Sigma. $N$-methyl pyrrolidone (NMP, >99.0%, 443778-500 ML) was purchased from Millipore–Sigma. The milling ball (Yttrium stabilized $ZrO_2$, 99.5%, $R = 5 \pm 0.3$ mm) was purchased from MTI Corporation. The 1 M $LiPF_6$ in a mixture of n a mixture of ethylene carbonate (EC): ethyl methyl Carbonate (EMC) ($V:V = 3:7$) electrolyte (battery grade, DJYLD2S-LB005-250) were purchased from LaborXing.

## FJH reaction

The FJH system was detailed in our previous publications[30,31]. A circuit diagram of the FJH setup and the FJH reaction box used in the experiments are shown in Supplementary Fig. 1 with essential safety precautions (Supplementary Note 1) for the FJH system[30]. The spent Li-ion batteries were discharged on a circuit until the voltage was below 2.5 V and then the electrodes were collected by manually disassembling the spent batteries. The cathode waste was used after directly removing it from the spent electrodes. Unless specified otherwise, the cathode materials and the conductive additive (10 wt% carbon black or 20 wt% spent anode graphite) were mixed evenly by grinding with a mortar and pestle for ~10 min. The reactants were loaded into a quartz tube with an inner diameter of 4 or 8 mm. The mass loads in 4- and 8-mm tube were 200 mg and 800 mg, respectively. Graphite rods and copper wool were used as electrodes and spacers, respectively. They were used to compress the reactants as shown in Fig. 1b. The graphite rods were in contact with the sample in the quartz tube. The electrical energy was provided by a capacitor bank composed of multiple aluminum electrolytic capacitors (6 mF, Mouser #80-PEH200YX460BQU2) in the circuit with a total capacitance of 60 mF (4-mm tube) or 132 mF (8-mm tube). The capacitor bank was charged by a D.C. supply that could reach 400 V. The flash duration was controlled by an Arduino controller relay in the circuit acting as a high-speed switch. Various cathode wastes, waste LCO, waste NMC, and cathode wastes combination of LCO and NMC were used to demonstrate the versatility of flash recycling method as listed in Table 1. After the FJH reaction, the reaction was permitted to cool for 3 min whereupon a commercial bar magnet with magnetic field strength ~5000 Oe was used to separate the ferromagnetic portion of the flash products. The mass ratio of the ferromagnetic portion was ~90 wt% and that of the nonmagnetic portion was ~10 wt%. The remaining ~10 wt% of flash product which was not captured by the magnet was collected and combined with minor portions from other FJH runs to be re-flashed, and the flash condition was the same as the one used for the primary flash. For the re-flash experiments, the small batch experiments (4-mm tube) were used as the demonstration. Thereby, ~60 wt% of re-flashed product can be magnetically recovered.

## Characterizations

The reactant and flash-recycled products were characterized through scanning electron microscopy (SEM) using a FEI Helios NanoLab 660 DualBeam SEM at 5 kV with a working distance of 4 mm. Transmission electron microscopy (TEM) images and selected area electron diffraction (SAED) patterns were taken with a JEOL 2100 F field emission gun transmission electron microscope at 200 kV. Atomic resolution high-resolution TEM (HR-TEM) and high-angle annular dark-field scanning transmission electron microscopy (HAADF-STEM) images were taken with FEI Titan Themis S/TEM instrument at 80 keV after accurate spherical aberration correction. X-ray photoelectron spectroscopy (XPS) data were collected with a PHI Quantera SXM Scanning X-ray Microprobe with a base pressure of $5 \times 10^{-9}$ Torr. Survey spectra were recorded using 0.5 eV step sizes with a pass energy of 140 eV. Elemental spectra were recorded using 0.1 eV step sizes with a pass energy of 26 eV. All the XPS spectra were corrected using the C $1s$ peaks (284.8 eV) as reference. For the depth analysis, an $Ar^+$ ion sputtering source was used to etch the surface layer. The average etching rate was calibrated and was ~7 nm min$^{-1}$ in the experiment which can be

further used to estimate the depth. X-ray diffraction (XRD) measurements were done by a Rigaku SmartLab Intelligent XRD system with filtered Cu Kα radiation ($\lambda = 1.5406$ Å). The reactants and flash products were analyzed on solid, dried samples using a Thermo Scientific Nicolet 6700 attenuated total reflectance Fourier transform infrared (ATR-FTIR) spectrometer (Waltham, MA). The metal contents in the magnetic products obtained from different cathode materials were quantified using a PerkinElmer Nexion 300 inductively coupled plasma mass spectroscopy (ICP-MS) and a PerkinElmer Optima 8300 inductively coupled inductively coupled plasma optical emission spectroscopy (ICP-OES) system. The samples were diluted with a 2% aqueous solution of nitric acid, and calibration curves were generated using 7 ICP standard solutions (blank solution, 1, 2, 5, 10, 25, and 50 ppm solutions), with the results used only from correlation coefficients that were greater than 0.999. For ICP-OES tests, the gas nebulizer flow rate range was set between 0.45 and 0.75 L min$^{-1}$, and 2 wavelengths per element were used in the axial mode unless otherwise stated: cobalt (228.616 and 230.786 nm), lithium (670.784 nm–radial mode–and 610.362 nm), nickel (231.604 and 341.476 nm) and manganese (257.610 and 259.372 nm). The magnetic characterizations were completed by a Quantum Design Magnetic Property Measurement System (MPMS). All the samples were ground into powder and then sealed into polytetrafluoroethylene (PTFE) tape. The tape was twisted into a small sphere and then transferred into a plastic straw for magnetic test. TGA was performed on a Mettler Toledo TGA/DSC 3+ system. TGA data were collected at a heating rate of 10 °C/min under air. The airflow was set to 80 mL/min.

## Cathode materials resynthesis

The cathode materials were resynthesized from the ferromagnetic FJH cathode products and in the context, they are named as flash-recycled cathode materials. ~1 g FJH cathode product was mixed with 0.2 g of lithium carbonate and then heated in a Mafu furnace (Carbolite RHF 1500). The sample temperature ramps to 800 °C with the heating rate of 20 °C min$^{-1}$ and is maintained at 800 °C for 12 h in the air. Afterward, the sample is allowed to cool to room temperature. The direct recycling method was also applied to prepare the resynthesized cathode, called direct recycled LCO. 1.0 g of waste LCO cathode was mixed with 0.2 g of lithium carbonate and then heated in a Mafu furnace (Carbolite RHF 1500). The sample temperature ramps to 800 °C with the heating rate of 20 °C min$^{-1}$ and is maintained at 800 °C for 12 h in the air. Afterward, the sample is allowed to cool to room temperature.

## Electrochemical tests

The resynthesized cathode material (areal capacity ~1.5 mAh cm$^2$) was used for the half-cell test. The cathode was prepared by grinding the mixture of resynthesized cathodes, conductive carbon black, and poly(vinyl difluoride) (PVDF) at a mass ratio of 8:1:1. A small amount (~2.5× of the total mass) of $N$-methyl pyrrolidone was used to form a homogeneous slurry. The slurries were formed by ball milling at 1500 rpm for 20 min. The cathode current collector was Al/C foil with a thickness of 18 μm. The slurry was applied to the Al/C foil by a doctor blade with blade spacing of 200 μm. The electrode was dried using a built-in heating cover placed on top of the electrode at 70 °C for 2 h and then put in a vacuum oven overnight. The temperature and pressure of the vacuum oven were set at 70 °C and ~10 mmHg. The area of the cathode was ~1.54 cm$^2$. The electrolyte used was 1 M $LiPF_6$ in a mixture of ethylene carbonate (EC): ethyl methyl carbonate (EMC) ($V: V = 3: 7$). The volume of the electrolyte in each coin cell was 30 μL. Lithium chips were used as the counter electrode with a polypropylene separator (~26 μm, SH416W14, SENIOR). Before the electrochemical test, the cells were pretreated at 0.05 C and 0.1 C between 3.0 and 4.2 V for 5 cycles, respectively. Subsequently, the cells were galvanostatically cycled between 3.0 and 4.2 V at 0.2 C for the stability tests.

CV voltammograms were taken with different scan rates from 0.1 mV s$^{-1}$ to 1 mV s$^{-1}$ in the range of 3.0–4.2 V using a CHI 680D electrochemical workstation. EIS measurements were conducted on the same electrochemical workstation by applying an alternating voltage of 5 mV in the frequency range from 0.01 Hz to 1 MHz.

## Sample digestion, leaching, and ICP-MS measurement

For all the cathode wastes and flash-recycled cathode materials, the contents of the battery metals, including lithium, cobalt, nickel, and manganese were measured. ~10 mg samples were digested in 5.0 mL *aqua regia* at 180 °C for 12 h. The *aqua regia* was prepared by mixing the nitric acid and hydrochloric acid in a molar ratio of ~1:3. The samples were filtered with PES membrane (0.22 μm) and diluted using HPLC plus grade water for ICP-MS measurement.

ICP-MS was conducted using a PerkinElmer Nexion 300 ICP-MS system. Mix 1 for ICP (10 mg L$^{-1}$, 10 wt% HNO$_3$, Millipore–Sigma) as the standard were purchased from Millipore–Sigma for the measurement of Li, Co, Ni, Mn, Al, Cu contents. The standard solutions were mixed and prepared at 1, 5, 10, 25, 50, 100, 1000 ppb. The sample concentration was calculated from the calibration curve.

## Atomistic calculations

Theoretical simulations were performed using first-principles density functional theory (DFT) calculations, as realized in the VASP software package. PAW potentials are employed for all species and the wave functions were expanded in a plane wave basis with an energy cutoff of 400 eV. All calculations are spin-polarized and employ the Perdew–Burke–Ernzerhof (PBE) exchange-correlation functional. Spin–orbit coupling was included in all the calculations. Rotationally invariant variant of the LSDA + U was employed. All structures underwent unrestrained structural relaxation until the forces on all atoms were less than 10$^{-3}$ eV/Å.

## Economic and environmental analysis

The GREET 2020 and EverBatt 2020 software, developed through Argonne National Laboratories, was used to estimate the cost and environmental impact in adopting different recycling processes. For comparison, the cathode materials derived from virgin sources were also analyzed. Our analysis was focused on the cumulative energy use, GHG production, and the potential net profit during the various recycling processes. More detailed discussion can be seen in Supplementary text.

## Reporting summary

Further information on research design is available in the Nature Portfolio Reporting Summary linked to this article.

# Data availability

The data supporting the findings of the study are available within the paper and its Supplementary Information. Source data are provided with this paper.

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

## Acknowledgements

The authors thank Dr. Bo Chen of Rice University for helpful input with the XPS results and Dr. Christopher Pennington for developing ICP-OES and ICP-MS methods. The funding of the research is provided by the Air Force Office of Scientific Research (FA9550-19-1-0296, J.M.T.), U.S. Army Corps of Engineers ERDC (W912HZ-21-2-0050, J.M.T.) and Rice Academy Fellowship (Y.C.). Theory and computations work is supported by the Department of Energy, BES (DE-SC0012547, K.V.B. and B.I.Y.). Permission to publish was granted by the Director, Geotechnical & Structures Laboratory, ERDC. The characterization equipment used in this project is partly from the Shared Equipment Authority (SEA) at Rice University.

## Author contributions

W.C. and J.M.T. conceived of the FJH process for cathode recycling. W.C., Y.C., and J.C. conducted the synthesis and characterizations with the help of C.G. R.V.S. helped with the resynthesis of the cathode materials and discussions on the battery testing. J.T.L. conducted the

SEM and EDS. D.X.L. aided with the temperature measurement and C.K. helped with the sample digestion before ICP-OES. Y.C. and J.C. conducted the ICP-MS test with the assistance of Z.W. E.A.M. conducted some TGA tests. G.G. and Y.H. assisted with the TEM and HAADF-STEM. B.D. offered useful suggestions. K.V.B. under the direction of B.I.Y. performed the atomistic modeling and wrote that section of the work. W.C., Y.C., J.C., R.V.S., and J.M.T. wrote the manuscript. All aspects of the research were overseen by J.M.T. All authors discussed the results and commented on the manuscript.

## Competing interests

Rice University owns intellectual property on the flash recycling process disclosed here. That intellectual property is licensed to a company in which JMT is a shareholder, but he is not an officer, director or employee. Conflicts of interest are mitigated through compliance with the Rice University Office of Research Integrity. The other authors claim no current conflicts of interest.
