## [Peer Review File · Nature Communications]

Nondestructive flash cathode recyclingREVIEWER COMMENTS

Reviewer #1 (Remarks to the Author):

In this manuscript, the authors report the recycling of end-of-life Li-ion batteries via flash recycling method, which involves a flash Joule heating (FJH) process combined with magnetic separation to restore fresh cathodes from waste cathodes. Relithiated cathodes show comparable electrochemical performance to new commercial counterparts. It is worth noting that the authors have conducted plenty of researches on flash Joule heating method for material synthesis and waste recycling. The innovations highlighted in this work are similar with their reported studies. Besides, many contributions have been devoted to recycle electrode materials of Li-ion batteries through flash Joule heating technique, such as graphite anode (EcoMat, 2022, 4(5): e12212; Advanced Materials, 2023, 35(8): 2207303; Nano Research, 2023, 16(4): 4240-4245; Adv. Funct. Mater. 2023, 33, 2302951) and cathode (ACS Energy Letters, 2023, 8: 3005-3012; PNAS, 2022, 119(20): e2202202119). The recycling mechanism is same, and no additional or enough innovations are presented in this work. Therefore, I recommend the rejection of this work for publication in Nature Communications. There are several concerns for the authors to consider, before submitting to other journals.

(1) The authors claim that "the other components, such as binder, cathode electrolyte interphase (CEI), conductive carbon and metal impurities, are either decomposed or magnetically separated in the flash recycling method", more experimental evidences are needed to confirm this. All metal impurities have magnetism? What's the existing form of conductive carbon? Decomposed?

(2) This work evaluated the recovery efficiencies of flash recycled cathodes by ICP-MS to determine the content of metal species. How about the purities of the recycled cathodes? which is also important, not just considering the metal species.

(3) What are the main reasons or issues for degraded performances of waste cathodes? And how flash joule heating method addresses these issues? More investigations and discussions should be devoted to the structure evolution from waste cathodes to flash recycled cathodes. The behind structure-activity relationship needs to be established.

Reviewer #2 (Remarks to the Author):

This work presents a novel economically viable method of recycling lithium ion battery cathode materials. The recovered cathode materials show promising potential but additional studies are need to determine if the material is battery grade. Specifically, there is little to no mention of impurities. Impurity removal is one of the key advantages claimed and the purities mentioned Al/Cu both are greater than 20ppm in the recovered product are not inline with industry standard <10ppm. Magnet impurity removal is known in industry to be ineffective at reducing the total magnet content to battery grade (<50ppb). This is one of the reasons chemical purification is required. If magnet impurity removal is going to be used total magnet content should be reported.

Electrochemical results should also be compared to a control sample as the electrochemical properties of commercial LCO are highly variable. 142mAh/g is a very good result for a standard undoped gen 1 LCO product. But there are many LCO materials capable of achieving much higher capacities and it is unclear what the starting material LCO in this studies capacity is. The same goes for cycle life. The value to comparing the proposed technology to traditional direct recycling approaches is limited as those are known to not be commercially viable.

There are also so non-trivial differences in D50, D10,D90 in the virgin vs recovered LCO materials and additional analysis/explanation would be helpful.

Reviewer #3 (Remarks to the Author):

The flash Joule heating (FJH) represents an advancement grounded in the reduction roasting of pyrometallurgy. This innovative technique achieves in-situ rapid reactions (150-300 ms), and the resulting flashed recycled products demonstrate electrochemical performance comparable to that of commercial LCO following re-lithiation. While the topic is inherently intriguing, several enhancements can be introduced in response to the following critiques:

1. LCO cathode active materials are typically regarded as non-ferromagnetic substances, impeding their easy separation from Al or Cu through magnetic separation. The author should elucidate the magnetic response exhibited by FJH flashed products by providing a comprehensive composition analysis of these products.
2. The reaction temperature in FJH is approximately 2500K. The generation of LiAlO_2 at 500C from the Al current collector is anticipated, potentially diminishing the Li recovery rate. Building upon Comment #1, a more thorough understanding is required regarding FJH flashed products to address this concern.
3. It is imperative to disclose the recovery rate for a single batch reaction and the number of re-flashing processes undertaken to achieve optimal performance.
4. The environmental and economic benefits of FJH in comparison to direct recycling are not evident. The authors are encouraged to reassess their claims in the abstract.
5. On Page 4, Line 90, carbon additives serve not only as conductive materials but also as reductants. This multifaceted role should be explicitly acknowledged.
6. Additional clarification is needed concerning why the diffusion coefficient derived from flashed recycling is higher than that from directly recycled LCO.
7. In terms of Life Cycle Assessment (LCA), the authors should expound upon the assumptions made regarding multiple re-flashing processes. A detailed explanation of how environmental impacts are allocated and managed in the LCA would enhance the transparency and credibility of the study.

Response to the reviewers

Reviewer #1:

In this manuscript, the authors report the recycling of end-of-life Li-ion batteries via flash recycling method, which involves a flash Joule heating (FJH) process combined with magnetic separation to restore fresh cathodes from waste cathodes. Relithiated cathodes show comparable electrochemical performance to new commercial counterparts. It is worth noting that the authors have conducted plenty of researches on flash Joule heating method for material synthesis and waste recycling. The innovations highlighted in this work are similar with their reported studies. Besides, many contributions have been devoted to recycle electrode materials of Li-ion batteries through flash Joule heating technique, such as graphite anode (EcoMat, 2022, 4(5): e12212; Advanced Materials, 2023, 35(8): 2207303; Nano Research, 2023, 16(4): 4240-4245; Adv. Funct. Mater. 2023, 33, 2302951) and cathode (ACS Energy Letters, 2023, 8: 3005-3012; PNAS, 2022, 119(20): e2202202119). The recycling mechanism is same, and no additional or enough innovations are presented in this work. Therefore, I recommend the rejection of this work for publication in Nature Communications. There are several concerns for the authors to consider, before submitting to other journals.

Thank you for the revision suggestions. However, we want to point out the failure mechanism and then highlight the innovation in the manuscript by comparing the current work and those previous contributions as mentioned by the reviewer.

First, we want to discuss the failure mechanism of the cathode materials, since the effectiveness of the recycling methods relies on how to solve these failures. The repeated cycling can cause the continuous electrolyte decomposition and CEI accumulation, degradation of binder particles, loss of conductive contact between cathode particles, irreversible Li inventory loss, phase change, dissolution of Cu and Al from current collector followed by migration towards electrodes, and so on.^{1,2} And the extent of these failures can also affect the electrochemical performance of the regenerated cathode materials.

Flash recycling method includes (1) flash Joule heating treatment, (2) magnetic separation and (3) solid-state relithiation as shown in Fig. 1a. We found that flash Joule heating treatment can trigger the carbothermal reaction occurring at the surface of the particles due to the hot spot effect we discussed in the manuscript, leading to the surface structure change of the cathode particles and formation of ferromagnetism. Besides, the CEI and binder will be thermally decomposed, attributing to the deagglomeration of the spent cathode particles. And the reduction and evaporation of metal impurities, such as Cu and Al, will occur during the FJH process. The subsequent magnetic separation process removes the other nonmagnetic portions, such as conductive carbon and metal impurities. The final relithiation process can replenish the Li inventory and reconstruct the cathode structure. Therefore, flash recycling method can replenish the Li inventory, reconstruct the surface structure, and improve crystallinity of cathode particles, which are important for non-destructive cycling method. In addition, the removal of metal impurities, such as Cu and Al, and deagglomeration of the cathode particles can also be achieved. These factors are also important to determine the electrochemical properties of recycled cathode materials, but they can be hard to solve using the traditional direct recycling methods. Based on our

summary here, the flash recycling method achieves the nondestructive recycling of spent cathode and flash recycled cathode demonstrates electrochemical performance comparable to the commercial counterpart.

Second, we also realize these previous contributions as mentioned by the reviewer. Since our work focused on cathode recycling, the comparison below will focus on the cathode recycling part. The *PNAS* paper³ used rapid thermal radiation method to convert spent $\text{LiNi}_{1-x-y}\text{Mn}_x\text{Co}_y\text{O}_2$ cathodes to bifunctional NiMnCo catalysts. But our main goal in the manuscript is to achieve nondestructive cathode recycling. The *ACS Energy Lett.* paper⁴ adopted the rapid Joule heating method to lithiate the spent cathode. To collect the spent cathode, they used base leaching, furnace annealing at 600 °C for 6 h and subsequent ball mixing of spent cathode with lithium carbonate in alcohol for 6 h. In comparison, our work highlights the FJH process to remove the impurities and deagglomerate spent cathode particles. Therefore, the flash recycling method is solvent- and water-free to achieve nondestructive cathode recycling.

The other four papers (*EcoMat*, 2022, 4(5): e12212; *Advanced Materials*, 2023, 35(8): 2207303; *Nano Research*, 2023, 16(4): 4240-4245; *Adv. Funct. Mater.* 2023, 33, 2302951) mentioned the utilization of flash Joule heating or carbothermal shock methods to achieve the effective regeneration of the graphite anode, which indicates that the methods we mentioned in the manuscript can solve both cathode and anode recycling problems arising from spent battery accumulation.

Based on the above analysis, we hope the reviewer is convinced that the flash recycling mechanism is different and there are enough innovations presented in our work. We hope therefore that the reviewer will reconsider our manuscript as suitable for *Nature Communications*.

1. J. P. Pender, G. Jha, D. H. Youn, J. M. Ziegler, I. Andoni, E. J. Choi, A. Heller, B. S. Dunn, P. S. Weiss, R. M. Penner, C. B. Mullins. Electrode degradation in lithium-ion batteries. *ACS Nano* **14**, 1243–1295 (2020).
2. P. Xu, Q. Dai, H. Gao, H. Liu, M. Zhang, M. Li, Y. Chen, K. An, Y. S. Meng, P. Liu, Y. Li, J. S. Spangenberg, L. Gaines, J. Lu, Z. Chen, Efficient direct recycling of lithium-ion battery cathodes by targeted healing. *Joule* **4**, 2609-2626 (2020).
3. M. Jiao, Q. Zhang, C. Ye, Z. Liu, X. Zhong, J. Wang, C. Li, L. Dai, G. Zhou, H. -M. Cheng, Recycling spent $\text{LiNi}_{1-x-y}\text{Mn}_x\text{Co}_y\text{O}_2$ cathodes to bifunctional NiMnCo catalysts for zinc-air batteries. *Proc. Natl. Acad. Sci.* **119**, e2202202119 (2022).
4. Y. -C. Yin, C. Li, X. Hu, D. Zuo, L. Yang, L. Zhou, J. Yang, J. Wan, Rapid, direct regeneration of spent LiCoO_2 cathodes for Li-ion batteries. *ACS Energy Lett.* **8**, 3005-3012 (2023).

Accordingly, we add the discussion and relevant references to the introduction to help the readers understand the advantages of this flash recycling method and the difference between this work and previous literature.

(P3) “...reagents. Recently, a rapid thermal radiation method has also been reported to convert spent cathodes into metal/metal oxide core-shell catalysts¹⁵. Traditional destructive recycling strategies...”

15. M. Jiao, Q. Zhang, C. Ye, Z. Liu, X. Zhong, J. Wang, C. Li, L. Dai, G. Zhou, H. -M. Cheng, Recycling spent $\text{LiNi}_{1-x-y}\text{Mn}_x\text{Co}_y\text{O}_2$ cathodes to bifunctional NiMnCo catalysts for zinc-air batteries. *Proc. Natl. Acad. Sci.* **119**, e2202202119 (2022).

(P3) “...The unique crystal structures of the cathode materials are as important as their chemical constitutions, as shown in the structure value estimation (Supplementary Table 1 and Supplementary Note 2)^{17,18}. In addition, the common failure mechanisms related to the cathode materials include continuous electrolyte decomposition and cathode electrolyte interphase (CEI) accumulation, degradation of binder particles, loss of conductive contact between cathode particles, irreversible Li inventory loss, surface structure change, and dissolution of Cu and Al from current collector followed by migration towards electrodes.^{16,19} Therefore, cathode healing through nondestructive strategies, focusing on how to effectively solve these failures, has recently gained more attention for battery recycling...”

19. J. P. Pender, G. Jha, D. H. Youn, J. M. Ziegler, I. Andoni, E. J. Choi, A. Heller, B. S. Dunn, P. S. Weiss, R. M. Penner, C. B. Mullins. Electrode degradation in lithium-ion batteries. *ACS Nano* **14**, 1243–1295 (2020).

(P3) “...Recently, an ultrafast repairing method using carbothermal shock method has also been proposed to replenish the Li inventory and reconstruct the surface structure²⁷. However, the performance...”

27. Y. -C. Yin, C. Li, X. Hu, D. Zuo, L. Yang, L. Zhou, J. Yang, J. Wan, Rapid, direct regeneration of spent LiCoO_2 cathodes for Li-ion batteries. *ACS Energy Lett.* **8**, 3005-3012 (2023).

(P4) “...The other components, such as binder, CEI, conductive carbon and metal impurities, are either decomposed or magnetically separated in the flash recycling method. This surface reaction facilitates the deagglomeration of the used cathode particles, which is beneficial for subsequent solid-state relithiation...”

(P16) “...In addition, recent works have demonstrated that the same FJH process and carbothermal shock method can be used to achieve the effective regeneration of the graphite anode from spent graphite^{34,50,67,68}, indicating that flash recycling method can simultaneously solve both cathode and anode recycling problems arising from spent battery accumulation...”

34. W. Chen, R. V. Salvatierra, J. T. Li, C. Kittrell, J. L. Beckham, K. M. Wyss, N. La, P. E. Savas, C. Ge, P. A. Advincula, P. Scotland, L. Eddy, B. Deng, Z. Yuan, J. M. Tour, Flash recycling of graphite anodes. *Adv. Mater.* **35**, 2207303 (2023).

50. T. Li, L. Tao, L. Xu, T. Meng, B. C. Clifford, S. Li, X. Zhao, J. Rao, F. Lin, L. Hu, Direct and rapid high-temperature upcycling of degraded graphite. *Adv. Funct. Mater.* 33, 2302951 (2023).
67. S. Dong, Y. Song, K. Ye, J. Yan, G. Wang, K. Zhu, D. Cao, Ultra-fast, low-cost, and green regeneration of graphite anode using flash joule heating method. *EcoMat.* 4, e12212 (2022).
68. J. Luo, J. Zhang, Z. Guo, Z. Liu, S. Dou, W. -D. Liu, Y. Chen, W. Hu, Recycle spent graphite to defect-engineered, high-power graphite anode. *Nano Res.* 16, 4240-4245 (2023).

(1) The authors claim that “the other components, such as binder, cathode electrolyte interphase (CEI), conductive carbon and metal impurities, are either decomposed or magnetically separated in the flash recycling method”, more experimental evidences are needed to confirm this. All metal impurities have magnetism? What’s the existing form of conductive carbon? Decomposed?

Thank you for the comments. TGA can be used to evaluate the weight loss from binder, CEI and conductive carbon based on their thermal stability. We have carried out the TGA for spent LCO before FJH treatment (waste LCO), after FJH treatment and magnetic separation (FJH-LCO), and after the final relithiation process (flash recycled LCO) as shown in Figure R1.

For waste LCO, we can see the weight loss at three different stages:

- (1) from 25 to ~250 °C, the weight loss is ~0.85%, which can result from the residual salt decomposition and CEI decomposition
- (2) from 250 to ~550 °C, the weight loss is ~4.48%, resulting from binder decomposition and conductive carbon oxidation.¹
- (3) from 550 to 1000 °C, the weight loss is ~2.01%, which can come from the thermal decomposition of inorganic lithium salts, such as Li₂CO₃, and spent cathode.

- 1 R. Zhan, T. Payne, T. Leftwich, K. Perrine, L. Pan, De-agglomeration of cathode composites for direct recycling of Li-ion batteries. *Waste Manag.* 105, 39-48 (2020).

For FJH-LCO, we can also see the weight change at three different stages:

- (1) from 25 to ~550 °C, the weight loss is ~1.40%, which can result from the carbon oxidation, mostly the carbon coating layer together with the cathode particles.
- (2) from 550 to ~820 °C, the weight increases ~4.72%, mainly resulting from the metal oxidation, such as Co and CoO to Co₃O₄.
- (3) from 820 to 1000 °C, the weight loss is ~3.95%, which can come from thermal decomposition of inorganic lithium salts, such as Li₂CO₃, and spent cathode after FJH treatment.

For flash recycled LCO, there is no obvious weight loss from 25 to ~250 °C and from 250 to ~550 °C, which shows a similar curve as the commercial LCO. In addition, we have characterized the structures of FJH-LCO, and flash recycled LCO using TEM as shown in Figure R2, and we can see that there is no CEI on the surface of flash recycled LCO. The

above result indicates that CEI, binder and conductive carbon can be removed after the flash recycling method.

Fig. R1. TGA results of various cathode materials. **a**, Waste LCO. **b**, FJH-LCO. **c**, Flash recycled LCO. The slight weight gain may be due to oxidation. **d**, Commercial LCO. TGA data were collected at a heating rate of 10 °C/min under air. The air flow was set to 80 mL/min. (Supplementary Figure 8)

Fig. R2. TEM results of various cathode materials. a, FJH-LCO. b, Flash recycled LCO. Inset is the fast Fourier transform analysis of the indicated line. (Supplementary Figure 24, panels d,g)

Fig. R3. ICP-MS results of various cathode materials. a, The concentration of impurity metals in waste LCO and FJH-LCO. b, The concentration of battery metals and impurity metals in flash recycling LCO and direct recycling LCO. The error bars reflect the standard deviations from at least three individual measurements. (Figure 2e, and Supplementary Figure 12)

For metal impurities, such as Cu and Al, we have characterized their contents using ICP-MS as shown in Figure R3. we can see that after FJH treatment and magnetic separation, the total Cu and Al contents are 8.2 and 90.9 ppm, respectively. We also determined the metal contents after the solid-state relithiation as shown in Figure R3. The total Cu and Al contents are 4.6 and 75.1 ppm, respectively for flash recycled LCO. The above values indicate that flash recycling method effectively decreases the total Cu and Al contents by ~75% and ~66%, respectively.

Fig. R4. Magnetic response of various cathode materials. **a**, The room temperature (300 K) hysteresis loops for FJH-LCO (orange curve) and the flash recycled LCO (red curve). **b**, The behavior of the hysteresis loop around the origin for FJH-LCO (orange curve) and the flash recycled LCO (red curve). **c**, The room temperature (300 K) hysteresis loops for FJH-LCO (orange curve) and commercial LCO (black curve). **d**, The behavior of the hysteresis loop around the origin for FJH-LCO (orange curve) and commercial LCO (black curve). (Supplementary Figure 25a,b and Supplementary Figure 7c,d)

The above-mentioned impurities, binder, CEI, conductive carbon and metal impurities, such as Cu and Al, show no ferromagnetic response, and therefore we can use the magnetic separation to collect the ferromagnetic portion of the cathode materials after FJH treatment, such as FJH-LCO and after the subsequent solid-state relithiation, the final product is not ferromagnetic as shown in Fig. R4.

Most of conductive carbon was removed after the magnetic separation, and the remaining carbon will be oxidized during the subsequent relithiation treatment.

Accordingly, we added the discussion to the manuscript and Supplementary Information to explain our statement. And we modified and added Figs. R1-R4 to the Supplementary Information.

(P6) “...At the high temperature of ~2500 K, the metal impurities were evaporated, while the other inert impurities such as binder and CEI, were decomposed, and most nonmagnetic conductive carbon was removed after the subsequent magnetic separation, as confirmed by the thermogravimetric analysis (TGA) results as shown in Supplementary Fig. 8 ...”

(P8) “These metal contents can be further reduced after the solid state relithiation step as shown in Supplementary Fig. 12. For flash recycled LCO, the contents of Cu and Al contents are 4.6 and 75.1 ppm, respectively, indicating effective decreases of Cu and Al contents by ~75% and ~66%, respectively...”

(2) This work evaluated the recovery efficiencies of flash recycled cathodes by ICP-MS to determine the content of metal species. How about the purities of the recycled cathodes? which is also important, not just considering the metal species.

Thank you for the comments. The purity of the cathode material is calculated to be ~99.74%, which meets the standard for battery grade cathode materials (>99.5%). Detailed discussion can be seen below.

We have used TGA, a magnetic property measurement system (MPMS), and ICP-MS to analyze the purities of recycled cathode materials, since ICP-MS can distinguish the metal impurities, such as Cu and Al, which are the major elements for anode and cathode current collectors. MPMS can distinguish magnetic impurities. TGA can be used to measure the other inert impurities such as conductive carbon, CEI and binder.

For the recycled cathode materials prepared by flash recycling method, the total contents of Cu and Al contents are 4.6 and 75.1 ppm, respectively, as shown in Figure R5, corresponding to notable content decreases by ~75% and ~66% for Al and Cu, respectively. In addition, from TGA analysis, there is no obvious weight loss from 25 to ~250 °C (~0.18%) and from 250 to ~550 °C (~0.07%) as shown in Figure R6, which shows a similar curve as the commercial LCO.

Fig. R5. ICP-MS results of recycled cathode materials. The error bars reflect the standard deviations from at least three individual measurements. (Supplementary Figure 12)

Fig. R6. TGA results of resynthesized cathode materials. Flash recycled LCO. The slight weight gain may be due to oxidation. TGA data were collected at a heating rate of 10 °C/min under air. The air flow was set to 80 mL/min. (Supplementary Figure 8, panel c)

We also analyzed the magnetic impurities using a magnetic property measurement system (MPMS). And the magnetic response of the FJH-LCO, flash recycled LCO and commercial LCO can be seen in Fig. R4. The commercial LCO is nonmagnetic without obvious magnetic impurities, and the moment is close to 0 even in a large magnetic field. The FJH-LCO is ferromagnetic, and the magnetic moment becomes nearly saturated at ~6000 Oe and the value reaches ~23.6 emu g⁻¹. Without the external magnetic field, the magnetic response was ~0.17 emu g⁻¹ for FJH-LCO. The flash recycled LCO is also not ferromagnetic and without obvious magnetic response. The magnetic moment is 6.6×10^{-6} emu g⁻¹ without external magnetic field. Since the total magnetic field for the Co₃O₄ is calculated to be ~70 emu g⁻¹ as shown in Fig. 3j, the magnetic content of final product flash recycled LCO is calculated to be ~94 ppb.

Therefore, considering all the above impurities, the purity of the cathode material is calculated to be ~99.74%, which meets the standard for battery grade cathode materials (>99.5%).

Fig. R4. Magnetic response of various cathode materials. **a**, The room temperature (300 K) hysteresis loops for FJH-LCO (orange curve) and the flash recycled LCO (red curve). **b**, The behavior of the hysteresis loop around the origin for FJH-LCO (orange curve) and the flash recycled LCO (red curve). **c**, The room temperature (300 K) hysteresis loops for FJH-LCO (orange curve) and commercial LCO (black curve). **d**, The behavior of the hysteresis loop around the origin for FJH-LCO (orange curve) and commercial LCO (black curve). (Supplementary Figure 25a,b and Supplementary Figure 7c,d)

Accordingly, we added this discussion to the manuscript and Supplementary Information. And we modified and add Fig. R4, R5 and R6 to the Supplementary Information.

(P8) “These metal contents can be further reduced after the solid-state relithiation step as shown in Supplementary Fig. 12. For flash recycled LCO, the contents of Cu and Al contents are 4.6 and 75.1 ppm, respectively, indicating effective decreases of Cu and Al contents by ~75% and ~66%, respectively...”

(P12) “...The LCO from flash recycling shows negligible mass loss (~0.25 wt%) from 298 to 823 K (Supplementary Fig. 8). The other magnet impurity contents are also analyzed using a magnetic property measurement system (MPMS) as shown in Supplementary Fig. 25, since the magnetic impurities can accelerate the self-discharge of the cathode materials⁵⁷. Based on the magnetic moment without external field, the total magnetic content of flash recycled LCO is estimated to be ~94 ppb. This value is close to the industrial standard of <50 ppb for battery grade materials and further optimization can be helpful to meet the standard during the industrialization and commercialization stage. Considering all the impurities, as characterized by TGA, ICP-MS and MPMS, the purities of flash recycled LCO is ~99.74%, which meets the standard for battery grade cathode materials of >99.5%. The TGA results

show no obvious mass loss (~0.28 wt%) until 1273 K, which reflects its high thermal stability and is comparable to the new commercial LCO...”

57. K. -P. Huang, G. T. -K. Fey, Y. -C. Lin, P, -J. Wu, J. -K. Chang, H. -M. Kao, Magnetic impurity effects on self-discharge capacity, cycle performance, and rate capability of LiFePO₄/C composites. *J Solid State Electrochem.* 21, 1767–1775 (2017).

(3) *What are the main reasons or issues for degraded performances of waste cathodes? And how flash joule heating method addresses these issues? More investigations and discussions should be devoted to the structure evolution from waste cathodes to flash recycled cathodes. The behind structure-activity relationship needs to be established.*

Thank you for the comments. We first discuss the failure mechanism of the cathode materials since the effectiveness of the recycling methods relies on how to solve these failures. The repeated cycling causes the continuous electrolyte decomposition and CEI accumulation, degradation of binder particles, loss of conductive contact between cathode particles, irreversible Li inventory loss, phase change, and dissolution of Cu and Al from the current collector followed by migration towards electrodes.^{1,2} The extent of these failures also affects the electrochemical performance of regenerated cathode materials.

1. J. P. Pender, G. Jha, D. H. Youn, J. M. Ziegler, I. Andoni, E. J. Choi, A. Heller, B. S. Dunn, P. S. Weiss, R. M. Penner, C. B. Mullins. Electrode degradation in lithium-ion batteries. *ACS Nano* 14, 1243–1295 (2020).
2. P. Xu, Q. Dai, H. Gao, H. Liu, M. Zhang, M. Li, Y. Chen, K. An, Y. S. Meng, P. Liu, Y. Li, J. S. Spangenberg, L. Gaines, J. Lu, Z. Chen, Efficient direct recycling of lithium-ion battery cathodes by targeted healing. *Joule* 4, 2609-2626 (2020).

We found that flash recycling method replenishes the Li inventory, reconstructs the surface structure, and improves crystallinity of cathode particles, which are important for the non-destructive cycling method. In addition, the removal of metal impurities, such as Cu and Al, and deagglomeration of the cathode particles is also achieved. These factors are also important to determine the electrochemical properties of recycled cathode materials, but they can be hard to solve using the traditional direct recycling methods. Due to the hot spot effect as we discussed in the manuscript, the carbothermal reaction will occur at the surface of the particles. The CEI and binder will be thermally decomposed. The reduction and evaporation of metal impurities will occur during the FJH process. The subsequent magnetic separation process removes the other nonmagnetic portion, such as conductive carbon. The final relithiation process replenishes the Li inventory and reconstructs the cathode structure. Therefore, the flash recycling method achieves the nondestructive recycling of spent cathodes and the flash recycled cathode demonstrates electrochemical performance comparable to the commercial counterpart.

For the impurities, we considered both the common metal impurities and other impurities. To measure the metal impurities, such as Cu and Al before and after FJH treatment, we have characterized the contents using ICP-MS as shown in Fig. R3 and Fig. 2f. We can see the content reduction for different metal impurities. For the change of other impurities, such as

binder, CEI, conductive carbon, we characterize the samples using TGA, SEM and TEM. The reduction of conductive carbon and removal of binder and CEI can be seen by TGA results as shown in Fig. R1 as discussed above in comment #1.

The de-agglomeration of cathode particles as seen by SEM images in Fig. R7h and Fig. 4a, and the absence of CEI as seen by HR-TEM images in Fig. R7g also confirmed the TGA results. The remaining conductive carbon is removed during the solid-state relithiation, since there is no obvious mass loss for flash recycle LCO as we can see in Fig. R6 above.

Fig. R7. Structural characterization of cathode materials at different stages during flash recycling. a-c, Waste LCO. d-f, FJH-LCO. g-i, Flash recycled LCO. For g, the inset is the fast Fourier transform analysis of the indicated line. (now Supplementary Figure 24)

For the structure evolution of the cathode particles in each step, we have characterized the structure using TEM, SEM and XRD as shown in Fig. R7 above. As we can see from the HR-TEM images in Fig. R7, the surface degradation, and the existence of the spinel structure on the surface of the waste LCO particles can be seen. After the FJH treatment and magnetic separation, the layered bulk structures can still be observed from XRD results in Fig. R7f. HR-TEM images as shown in Fig. R7d also show the existence of the layered structures, which is consistent with the XRD results. The ferromagnetic Co_3O_4 with spinel structure is observed on the surface of the cathode particles after the FJH treatment, therefore, the

cathode material is magnetically separated after the FJH treatment. The subsequent solid-state relithiation converts the FJH cathodes to the resynthesized cathode materials. We also characterize the structure using TEM, SEM and XRD, as shown in Fig. R7. The reformation of the layered structure at the surface and the existence of individual cathode particles indicates effective resynthesis of the cathode materials using the flash recycling method.

Accordingly, we added discussion to the manuscript and Supplementary Information. And we modified and added Fig. R7 to the Supplementary Information.

(P3) “...The unique crystal structures of the cathode materials are essential, as are their chemical constitutions, as shown in the structure value estimation (Supplementary Table 1 and Supplementary Note 2)^{17,18}. In addition, the common failure mechanisms related to the cathode materials include continuous electrolyte decomposition and cathode electrolyte interphase (CEI) accumulation, degradation of binder particles, loss of conductive contact between cathode particles, irreversible Li inventory loss, surface structure change, and dissolution of Cu and Al from current collector followed by migration towards electrodes.^{16,19} Therefore, cathode healing through nondestructive strategies, focusing on how to effectively solve these failures, has recently gained more attention for battery recycling...”

(P12) “...A summary of the structure evolution from waste LCO, to FJH-LCO and final flash recycled LCO is shown in Supplementary Fig. 24, which highlights the surface reaction features of FJH treatment and cathode structure reconstruction after the solid-state reaction...”

Reviewer #2:

This work presents a novel economically viable method of recycling lithium ion battery cathode materials. The recovered cathode materials show promising potential but additional studies are need to determine if the material is battery grade. Specifically, there is little to no mention of impurities. Impurity removal is one of the key advantages claimed and the purities mentioned Al/Cu both are greater than 20ppm in the recovered product are not in line with industry standard <10ppm. Magnet impurity removal is known in industry to be ineffective at reducing the total magnet content to battery grade (<50ppb). This is one of the reasons chemical purification is required. If magnet impurity removal is going to be used total magnet content should be reported.

Thank you for the comments. We first confirmed the quality of battery grade LCO cathode materials from commercial sources, and the purity requirement is no less than 99.5%, with the impurity Cu no more than 50 ppm (<https://www.samaterials.com/lithium-cobalt-oxide-cathode-material-lco-powder.html>; <https://www.mseshop.com/products/mse-pro-lithium-cobalt-oxide-licoo-sub-2-sub-lco-cathode-powder-500g?variant=7110339076>).

Indeed, we found that some vendors can provide more expensive cathode materials with better quality, like the reviewer mentioned, <10 ppm or even 1 ppm for Cu metal (<https://www.mtixtl.com/LiCoO2PowderforLi-ionbatteryCathode200g/bag-EQ-Lib-LCO-1.aspx>). None of them mentioned the Al content. We analyzed the impurities contents for the flash recycled LCO product using ICP-MS and found that the contents of Cu and Al contents are 4.6 and 75.1 ppm, respectively, as shown in Fig. R3 below, which meets the requirement of <10 ppm for Cu above.

Fig. R3. ICP-MS results of various cathode materials. **a**, The concentration of impurity metals in waste LCO and FJH-LCO. **b**, The concentration of battery metals and impurity metals in flash recycling LCO and direct recycling LCO. The error bars reflect the standard deviations from at least three individual measurements. (Figure 2e, Supplementary Figure 12)

Fig. R4. Magnetic response of various cathode materials. **a**, The room temperature (300 K) hysteresis loops for FJH-LCO (orange curve) and the flash recycled LCO (red curve). **b**, The behavior of the hysteresis loop around the origin for FJH-LCO (orange curve) and the flash recycled LCO (red curve). **c**, The room temperature (300 K) hysteresis loops for FJH-LCO

(orange curve) and commercial LCO (black curve). d, The behavior of the hysteresis loop around the origin for FJH-LCO (orange curve) and commercial LCO (black curve). (Supplementary Figure 25a,b and Supplementary Figure 7c,d)

Then, we analyzed the magnetic impurities using a magnetic property measurement system (MPMS). And the magnetic response of the FJH-LCO, flash recycled LCO and commercial LCO can be seen in Fig. R4. The commercial LCO is nonmagnetic without obvious magnetic impurities, and the moment is close to 0 even at a large magnetic field. The FJH-LCO is ferromagnetic, and the magnetic moment becomes nearly saturated at ~6000 Oe and the value reaches ~23.6 emu g⁻¹. Without the external magnetic field, the magnetic response was ~0.17 emu g⁻¹ for FJH-LCO. The flash recycled LCO is also not ferromagnetic and without obvious magnetic response. The magnetic moment is 6.6×10^{-6} emu g⁻¹ without external magnetic field. Since the total magnetic field for the Co₃O₄ is calculated to be ~70 emu g⁻¹ as shown in Fig. 3j, the magnetic content of FJH-LCO is calculated to be ~0.24% and for the final product flash recycled LCO, the value is ~94 ppb. We also use the same method in the previous literature⁵⁷, to screen the magnetic part of the flash recycled LCO, and no obvious powder can be attracted to the strong neodymium magnetic bar, as shown in Video R1.

Even through the magnetic content of the final product (~94 ppb) is slightly higher than the industrial standard of <50 ppb for battery grade materials, our results have demonstrated the effectiveness of flash recycling method to regenerate cathode materials. Further optimization can be carried out to bridge this gap and meet the industrial standard if this method is chosen to be industrialized or commercialized.

Accordingly, we add the discussion to the manuscript and Supplementary Information. And we modified and added Figs. R3 and R4 to the Supplementary Information.

(P8) “These metal contents can be further reduced after the solid-state relithiation step as shown in Supplementary Fig. 12. For flash recycled LCO, the contents of Cu and Al contents are 4.6 and 75.1 ppm, respectively, indicating effective decreases of Cu and Al contents by ~75% and ~66%, respectively...”

(P12) “...The LCO from flash recycling shows negligible mass loss (~0.25 wt%) from 298 to 823 K (Supplementary Fig. 8). The other magnetic impurity contents are also analyzed using a magnetic property measurement system (MPMS) as shown in Supplementary Fig. 25, since the magnetic impurities can accelerate the self-discharge of the cathode materials⁵⁷. Based on the magnetic moment without external field, the total magnetic content of flash recycled LCO is estimated to be ~94 ppb. This value is close to the industrial standard of <50 ppb for battery grade materials and further optimization can be helpful to meet the standard during the industrialization and commercialization stage. The magnet test also confirms that no cathode powder can be attracted (Supplementary Video 1) as suggested by the previous literature⁵⁷. Considering all the impurities, as characterized by TGA, ICP-MS and MPMS, the purities of flash recycled LCO is ~99.74%, which meets the standard for battery grade cathode materials of >99.5%. The TGA results show no obvious mass loss (~0.28 wt%) until 1273 K, which reflects its high thermal stability and is comparable to the new commercial LCO...”

57. K. -P. Huang, G. T. -K. Fey, Y. -C. Lin, P. -J. Wu, J. -K. Chang, H. -M. Kao, Magnetic impurity effects on self-discharge capacity, cycle performance, and rate capability of LiFePO_4/C composites. *J Solid State Electrochem.* **21**, 1767–1775 (2017).

Electrochemical results should also be compared to a control sample as the electrochemical properties of commercial LCO are highly variable. 142mAh/g is a very good result for a standard undoped gen 1 LCO product. But there are many LCO materials capable of achieving much higher capacities and it is unclear what the starting material LCO in this studies capacity is. The same goes for cycle life. The value to comparing the proposed technology to traditional direct recycling approaches is limited as those are known to not be commercially viable.

Thank you for the comments. We agree that the comparison between the control LCO samples from the same vendor and the resynthesized LCO samples can be very helpful to evaluate the flash recycling method. However, the LCO waste we used was from spent batteries (LG Chem 112711, B052R785-9005A) obtained from Lenovo laptop computers after several years usage. It would be difficult to get the same LCO sample now.

Fig. R8. Electrochemical performance of various cathode materials. a, Waste LCO, flash recycled LCO and commercial LCO. b, Resynthesized LCO using hydrometallurgical method. (Figure 4h and Supplementary Figure 30)

Therefore, in the following discussion, to demonstrate the effectiveness of the flash recycling method, we compared the electrochemical performance of resynthesized LCO from different methods, including the destructive hydrometallurgical method that can be commercially viable, and this flash recycling method as shown in the figure. We see similar specific capacity for the LCO synthesized from this flash recycling method and the hydrometallurgical method, indicating the effectiveness of flash recycling method.

On the other hand, the specific capacity of LCO is dependent on the cut-off voltage during the charging stage, for example, the capacity is 140 mAh g⁻¹ with 4.2 V cut-off voltage, while the value can reach 220 mAh g⁻¹ with 4.6 V cut-off voltage²¹. Therefore, since we used the same cut-off voltage for commercial and flash recycled LCO, the specific capacity can be somehow comparable.

21. J. Wang, K. Jia, J. Ma, Z. Liang, Z. Zhuang, Y. Zhao, B. Li, G. Zhou, H. -M. Cheng, Sustainable upcycling of spent LiCoO₂ to an ultra-stable battery cathode at high voltage. *Nat. Sustain.* **6**, 797-805 (2023).

Accordingly, we add the discussion in the manuscript and Supplementary Information. And we modify and add Figs. R8 to the Supplementary Information.

(P14) “...**The electrochemical performance of resynthesized LCO using commercially viable recycling methods, such as hydrometallurgical method⁹ is shown in Supplementary Fig. 30, and similar specific capacity can be obtained.** Therefore, the proposed flash recycling method is effective to regenerate the cathode materials from cathode waste....”

There are also so non-trivial differences in D50, D10, D90 in the virgin vs recovered LCO materials and additional analysis/explanation would be helpful.

Thank you for the comments. We have measured the sizes of 100 LCO particles for both commercial and flash recycled samples. The size distribution and the information for D10, D50 and D90 can be seen in the figure below: We see that commercial and flash recycled LCO have similar size distribution. D10, D50 and D90 are 6.1, 13.6 and 19.7 μm, respectively, for flash recycled LCO. D10, D50 and D90 are 5.7, 12.5 and 19.8 μm, respectively, for commercial LCO.

Fig. R9. SEM images of different cathode materials. a-b, Flash recycled LCO. c-d, Commercial LCO. (Supplementary Figure 21b, Figure 4a, Supplementary Figure 21c and 21d)

Fig. R10. The particle sizes of different cathode materials. The size distribution of a, flash recycled LCO and b, commercial LCO. c, The size below which 10%, 50% or 90% of all particles are found for flash recycled LCO and commercial LCO (D10, D50 and D90). (Supplementary Figure 22)

Accordingly, we added Figure R10 to the Supplementary Information and added discussion to the manuscript:

(P11) “...The size distribution is analyzed using SEM images (Supplementary Fig. 21) and the size below which 10%, 50% and 90% of all particles are found (D10, D50, D90) are compared between commercial LCO and flash recycled LCO (Supplementary Fig. 22). D10, D50 and D90 are 6.1, 13.6 and 19.7 μm , respectively, for flash recycled LCO, while these values are 5.7, 12.5 and 19.8 μm , respectively, for commercial LCO. Therefore, commercial and flash recycled LCO have similar size distributions....”

Reviewer #3:

The flash Joule heating (FJH) represents an advancement grounded in the reduction roasting of pyrometallurgy. This innovative technique achieves in-situ rapid reactions (150-300 ms), and the resulting flashed recycled products demonstrate electrochemical performance comparable to that of commercial LCO following re-lithiation. While the topic is inherently intriguing, several enhancements can be introduced in response to the following critiques:

Thank you for the positive comments. We have addressed the concerns point-by-point as shown below.

1. LCO cathode active materials are typically regarded as non-ferromagnetic substances, impeding their easy separation from Al or Cu through magnetic separation. The author should elucidate the magnetic response exhibited by FJH flashed products by providing a comprehensive composition analysis of these products.

Thank you for the comments. We have analyzed the composition of LCO after FJH treatment (FJH-LCO) using XPS as shown in the figure below. As we can see, after the FJH process, there was an obvious binding energy shift towards higher values for Co 2p spectra at the surface of the cathode, which indicated that the cathode surface was partially reduced from Co^{3+} to Co^{2+} . This result is consistent with our characterization and simulation in Figure 3.

In addition, the splitting of the O 1s spectra to O_α (~532.6 eV) and O_β (~530.3 eV), showed a transition from adsorbed oxygen species to lattice oxygen species³⁷, which confirmed the existence of lithiated metal oxides below the surface. There was one extra peak resulting from C=O at the surface, which was not observed below the surface. Combined with the Li 1s spectra and FTIR results in Supplementary Fig. 6, the data indicate that some lithium carbonate salt might have formed at the surface. Hence, the carbothermal reaction of the LCO cathode could occur during the FJH treatment, leading to the reduction of oxidation states for cobalt from +3 to +2 and formation of oxides, such as Co_3O_4 at the surface. The formation of Co_3O_4 can also be confirmed by the XRD result as shown in the figure below. The oxide was ferromagnetic as confirmed by magnetic response and simulation, and therefore the flashed cathode can be magnetically separated from Al and Cu.

37. X. Chen, C. Luo, J. Zhang, J. Kong, T. Zhou, Sustainable recovery of metals from spent lithium-ion batteries: A green process. *ACS Sustain. Chem. Eng.* **3, 3104-3113 (2015).**

Fig. R11. The high-resolution elemental analysis of new LCO after FJH treatment. a, C 1s. b, Co 2p. c, O 1s. d, XRD result. Powder diffraction file: 43-1003, Co₃O₄. 16-0427, LiCoO₂. (Supplementary Figure 17)

Accordingly, we added Figure R11 to the Supplementary Information and added a discussion to the manuscript:

(P9) “...The composition analysis of FJH-nLCO shows that there is a large binding energy shift towards higher value for Co 2p spectra at the surface of the cathode particles (Supplementary Fig. 17-18), indicating that the cathode surface was partially reduced from Co³⁺ to Co²⁺ and formation of ferromagnetic oxides during the FJH treatment⁹. The formation of Co₃O₄ is also confirmed by the XRD result of FJH-nLCO as shown in Supplementary Fig. 17d. Similar phenomena on oxidation state reduction are observed for other cathode chemistries, such as new NMC (Supplementary Fig. 19). The splitting of the O 1s spectra to O_α (~532.6 eV) and O_β (~530.3 eV), showed a transition from adsorbed oxygen species to lattice oxygen species³⁷, which confirmed the existence of lithiated metal oxides below the surface. There was one extra peak resulting from C=O at the surface, which was not observed below the surface. Combined with the Li 1s spectra and FTIR results in

Supplementary Fig. 6, it indicates that some lithium carbonate salt might form at the surface....”

2. The reaction temperature in FJH is approximately 2500 K. The generation of LiAlO_2 at 500 C from the Al current collector is anticipated, potentially diminishing the Li recovery rate. Building upon Comment #1, a more thorough understanding is required regarding FJH flashed products to address this concern.

Thank you for the comments. The deconvolution of XPS analysis of the flashed products is discussed in comment 1 above.

In addition, since the cathode waste was manually collected from a used electrode as we discussed in the Experimental Section, the introduction of Al can be reduced. Based on the ICP-MS results, the total concentration of Al was ~223.5 ppm as shown in Fig. R12.

Fig. R12. The concentration of impurity metals in waste LCO and FJH-LCO. The error bars reflect the standard deviations from at least three individual measurements. (panel e of Figure 2 in the manuscript)

Conversely, the XRD pattern of LiAlO_2 is not observed from the flashed products as shown in the figure below. Therefore, the effect of Li recovery rate due to the generation of LiAlO_2 can be negligible.

Fig. R13. The XRD results of various cathode materials before and after FJH treatment. **a,** Waste LCO. **b,** CW. **c,** NMC. **d,** FJH-LCO. **e,** FJH-CW. **f,** FJH-NMC. Powder diffraction file: 38-1464, LiAlO_2 . (Supplementary Figure 11)

Accordingly, we added Figure R13 to the Supplementary Information and added some discussion to the manuscript:

(P8) “...Since the total concentration of Al is low within the spent cathodes, there is no obvious formation of LiAlO_2 as confirmed by the XRD results (Supplementary Fig. 11)....”

3. It is imperative to disclose the recovery rate for a single batch reaction and the number of re-flashing processes undertaken to achieve optimal performance.

Thank you for the comments. We have carried out the flash experiment and explored the relationship between the flash pulses and the battery metal contents as shown in Figure R14 below. We measured the concentration of battery metals including both Li and Co within the nonmagnetic portion of the flashed products derived from waste LCO. From the curves it can be seen that as we increased the flash pulses from 1 to 3, the concentration of battery metals decreased from 1014.8 to 19.8 ppm for Co and from 99.8 to 11.1 ppm for Li, respectively, which indicates that the recovery yield for a single batch reaction increases slightly with an increase in flash pulses.

Fig. R14. The relationship between the flash pulses and the concentration of battery metals within the nonmagnetic portion of the flashed products derived from waste LCO. The error bars reflect the standard deviations from at least three individual measurements. **(Supplementary Figure 9)**

However, we want to point out that once the reaction parameters, such as flash voltage, capacitance, reaction duration, and reactant resistance, have been well-optimized, one flash reaction can be enough to obtain satisfied recovery yield for all the battery metals.

According to the above discussion, we add Figure R14 to the Supplementary Information and added discussion to the manuscript:

(P6) “...For FJH-LCO derived from waste LCO, the average recovery yields are ~94.2% for Co and ~96.3% for Li, respectively, (Fig. 2b) after just one flash treatment. To explore the relationship between the recovery yield and flash pulses for spent cathode with single component, such as waste LCO, the battery metal contents within the nonmagnetic portion are measured (Supplementary Fig. 9). As the flash pulses increase from 1 to 3, the concentration of battery metals decreased from 1014.8 to 19.8 ppm for Co and from 99.8 to 11.1 ppm for Li, respectively, which indicated that the recovery yield for a single batch reaction increased marginally as the increase of flash pulses. Therefore, one flash treatment is used for spent cathode with single component in the LCA calculation....”

4. *The environmental and economic benefits of FJH in comparison to direct recycling are not evident. The authors are encouraged to reassess their claims in the abstract.*

Thank you for the comments. We have made changes in the abstract and in the manuscript. We focused on the comparison of environmental and economic benefits between flash recycling and the traditional destructive recycling method in the abstract, and we established that these values are similar when compared to the direct recycling method in the manuscript. Therefore, flash recycling can be an alternative strategy for non-destructive recycling methods.

According to the above discussion, we modified the manuscript:

(P2) “...Life-cycle-analysis highlights that flash recycling has higher environmental and economic benefits over traditional **destructive** recycling processes...”

(P4) “...Life cycle assessment (LCA) comparisons to the present **destructive** cathode recycling methods demonstrate that...”

5. *On Page 4, Line 90, carbon additives serve not only as conductive materials but also as reductants. This multifaceted role should be explicitly acknowledged.*

Thank you for the comments. We have made the change in the manuscript to acknowledge the multifaceted role of carbon additives as shown below:

(P5) “The carbon additives can surround the cathode particles, **which not only bridges the inner circuit to increase the electrical conductivity of the mixture, but also serves as the reductant for the cathode waste.** Therefore, the current will mainly pass through the conductive carbon and the generated electrothermal energy will transfer from these hot spots to adjacent cathode particles, **contributing to the local carbothermal reduction at the surface of cathode particles.**”

6. *Additional clarification is needed concerning why the diffusion coefficient derived from flashed recycling is higher than that from directly recycled LCO.*

Thank you for the comments. We have found that the diffusion coefficient of flash recycled LCO is like that of the directly recycled LCO during the discharging stage, but this value of flash recycled LCO is ~1.8× higher during charging stage. As we can see from Figure R15, the main difference between these two cathode materials is from the peak current I_p when higher voltage scanning rates are applied, especially when the rate exceeds 0.8 mV s^{-1} . At a fast-scanning rate, the kinetic processes become more important. The polarization can come from these three different stages, (1) Li^+ diffusion from the bulk solution to the surface of the electrode; (2) Li^+ diffusion across the CEI on the cathode particles; (3) Li^+ intercalation and arrangement within the cathode particles. Since the electrolyte is the same for the test, the main difference arises from stages 2 and 3.

Fig. R15. Cyclic voltammetry results of various cathode materials at different voltage scanning rates. a, Flash recycled LCO. b, Waste LCO. c, Direct recycling LCO. d, Comparison of diffusion coefficients of Li^+ in direct recycling LCO and flash recycled LCO. I_p : peak current. (now Supplementary Figure 26)

Fig. R16. SEM images of different resynthesized cathode materials. a, Direct recycling LCO. b, Flash recycled LCO. (Supplementary Figure 21a,b)

From the SEM images, we found the existence of cathode particle agglomeration for direct recycling LCO, while each individual cathode particle can be observed for flash recycled LCO. Therefore, due to the severe agglomeration, the effective area between the cathode

particles and the electrolyte becomes smaller, which can slow down the Li^+ diffusion across the CEI on the cathode particles.

Fig. R17. XRD results of various cathode materials. a, Waste LCO. b, Commercial LCO. c, Direct recycling LCO. d, Flash recycling LCO. e, Intensity ratio between (003) and (104) peaks (structure factor) for various cathode materials. (Figure 4c and Supplementary Figure 23)

Conversely, agglomeration can also prevent effective relithiation and afford a lower value of structure factor, which is the intensity ratio between (003) and (104) peaks. This can be found for direct recycling LCO, indicating cation mixing between transition metals and lithium. Therefore, the Li^+ intercalation and arrangement within the cathode particles can also be affected at the high scanning rate.

Accordingly, we added the discussion above to the manuscript as shown below:

(P13) “...the diffusion coefficient of flash recycled LCO is recovered from waste LCO, and the value is $\sim 1.8\times$ higher compared to direct recycling LCO during the charging stage (Supplementary Fig. 26d), presumably arising from the complete de-agglomeration of the resynthesized cathode particles and better crystallinity with less cation mixing for flash recycled LCO...”

7. In terms of Life Cycle Assessment (LCA), the authors should expound upon the assumptions made regarding multiple re-flashing processes. A detailed explanation of how environmental impacts are allocated and managed in the LCA would enhance the transparency and credibility of the study.

Thank you for the comments. As we demonstrated in comment 3, we found that multiple flashes only contribute to a slight increase (~0.1%) for waste LCO. And for waste NMC, a single flash can also achieve >95.5% recovery yield as shown in Figure 2c. Multiple flash pulses will increase the energy input and GHG emission, while the net increase of recovery yield is not enough to compensate for the above cost. Therefore, we consider the LCA based on a single flash as a comparison in this work.

For the cathode waste with mixed chemistries, such as a waste LCO and NMC mixture, we indeed realized that re-flashing treatment can be very helpful to increase the recovery yields of battery metals. However, the LCA for the cathode waste with mixed chemistries can be very hard to estimate even for the pyrometallurgical and hydrometallurgical methods, due to the potential metal separation problem and subsequent resynthesis processes with mixed metal ratios since closed-loop recycling is considered. Therefore, this analysis is beyond our consideration. Our LCA is based on a flash treatment with simple cathode chemistry.

For environmental impacts, we considered the energy and waste consumption, and greenhouse gas emission for different recycling methods. Since the pyrometallurgical method, hydrometallurgical method and direct recycling method have been documented and can be found using Everbatt 2020 (ref. 17), we will discuss below how we estimated the environmental impact of the flash recycling method. We assume that 1 MJ electricity produces 0.13 kg GHG and 0.67 L water, which comes from GREET 2020 (ref. 16). The energy consumption will come from electricity usage during the cathode relithiation, FJH process, and spent battery disassembly.

Cathode relithiation requires sintering the cathode powder and extra lithium salt at 1073 K for 12 h, which is the same as the direct recycling method. A commercial furnace can be used here, and the parameters include temperature, power, and mass loading of ~1073 K, 40 kW and 245 kg, respectively. Therefore, the energy consumption is 7.05 MJ per kg cathode powder. The water consumption for cathode relithiation is based on the above relationship, which is $(7.05 \times 0.67) \text{ L} = 4.726 \text{ L}$. In addition to the GHG produced from using electricity, the decomposition of Li_2CO_3 salt can also produce CO_2 , which accounts for 0.118 kg GHG emission. Therefore, the total GHG emission for cathode relithiation is $(7.05 \times 0.13 + 0.118) \text{ kg} = 1.035 \text{ kg}$.

Similarly, the electricity usage for the FJH process is 0.31 kWh kg^{-1} ; here we consider 10% overhead as well for reaction atmosphere and the total energy is $(0.31 \times 3.6 \times 1.1) \text{ MJ} = 1.23 \text{ MJ}$. The water usage and GHG emission for FJH process is calculated based on the above relationship. Therefore, the water usage is 0.825 L and GHG emission is 0.160 kg, respectively.

The disassembly step can be achieved by a commercial core drill with a silicon carbide blade, and scrapping is calculated based on Everbatt 2020 (ref. 17). Since there is no extra water required and no extra GHG emission, the total water usage and GHG emission for spent battery disassembly is based on the energy consumption in this step. The electricity consumption is estimated to be $(29.828 + 149.14) \text{ kWh}$ for 1740 kg spent batteries,

corresponding to 0.38 MJ per kg spent batteries. Therefore, the water usage is 0.26 L and GHG emission is 0.050 kg, respectively.

The same discharge step is used to estimate the energy and waste consumption, and greenhouse gas emission as shown in direct recycling method. The average water assumption is 0.5 L per kg spent batteries. The energy consumption is estimated based on the usage of a conveyor, which is ~0.03 MJ per kg spent batteries. Therefore, the water usage is $(0.50 + 0.03 \times 0.67)$ L = 0.52 L and GHG emission is (0.03×0.13) kg = 0.004 kg, respectively. The other detailed step considerations and a table of values are listed in Supplementary Table 5 and Supplementary Note 1.

Accordingly, we added the above discussion about the flash recycling method to the Supplementary Note 1 on pp 7-8:

For environmental impacts, we considered the energy and waste consumption, and greenhouse gas emission for different recycling methods. We assume that 1 MJ electricity produces 0.13 kg GHG and 0.67 L water, which comes from GREET 2020 (ref. 17). The energy consumption will come from electricity usage during the cathode relithiation, FJH process, and spent battery disassembly.

Cathode relithiation requires sintering the cathode power and extra lithium salt at 1073 K for 12 h, which is the same as the direct recycling method. A commercial furnace can be used here, and the parameters include temperature, power, and mass loading of ~1073 K, 40 kW and 245 kg, respectively. Therefore, the energy consumption is 7.05 MJ per kg cathode power. The water consumption for cathode relithiation is based on the above relationship, which is (7.05×0.67) L = 4.726 L. In addition to the GHG produced from using electricity, the decomposition of Li_2CO_3 salt can also produce CO_2 , which accounts for 0.118 kg GHG emission. Therefore, the total GHG emission for cathode relithiation is $(7.05 \times 0.13 + 0.118)$ kg = 1.035 kg.

Similarly, the electricity usage for FJH process is 0.31 kWh kg^{-1} , here we consider 10% overhead as well for reaction atmosphere and the total energy is $(0.31 \times 3.6 \times 1.1)$ MJ = 1.23 MJ. The water usage and GHG emission for FJH process is calculated based on the above relationship. Therefore, the water usage is 0.825 L and GHG emission is 0.160 kg, respectively.

The disassembly step can be achieved by a commercial core drill with a silicon carbide blade, and scrapping is calculated based on Everbatt 2020 (ref. 18). Since there is no extra water required and no extra GHG emission, the energy consumption is used to estimate the water usage and GHG emission in this step. The electricity consumption is estimated to be $(29.828 + 149.14)$ kWh for 1740 kg spent batteries, corresponding to 0.38 MJ per kg spent batteries. Therefore, the water usage is 0.26 L and GHG emission is 0.050 kg, respectively.

The same discharge step is used to estimate the energy and waste consumption, and greenhouse gas emission as shown in other recycling methods. The average water assumption is 0.5 L per kg spent batteries. The energy consumption is estimated based on the usage of a conveyor, which is ~0.03 MJ per kg spent batteries. Therefore, the water usage is $(0.50 + 0.03 \times 0.67)$ L = 0.52 L and GHG emission is (0.03×0.13) kg = 0.004 kg, respectively.

-=-=-=-=-

With these changes, corrections and additions made at the request of the reviewers, we respectfully request that the revised manuscript be considered for publication.

Sincerely,

Jim

James M. Tour, Ph.D., FRSC, NAI, NAE

REVIEWERS' COMMENTS

Reviewer #1 (Remarks to the Author):

Considering the authors have detailedly claimed the innovations of this work and well addressed the comments, the revised manuscript could be accepted for publication.

Reviewer #2 (Remarks to the Author):

The edits made by the authors have sufficiently addressed the issues needed for publication.